# Effect of the Atlantic Meridional Overturning Circulation on Atmospheric pCO$_2$ Variations

Daan Boot[1], Anna S. Von Der Heydt[1,2], and Henk A. Dijkstra[1,2]

[1]Institute for Marine and Atmospheric research Utrecht, Department of Physics, Utrecht University, Utrecht, the Netherlands
[2]Center for Complex Systems Studies, Utrecht University, Utrecht, the Netherlands

**Correspondence:** D. Boot <d.boot@uu.nl>

**Abstract.** Proxy records show large variability of atmospheric pCO$_2$ on different time scales. Most often such variations are attributed to a forced response of the carbon cycle to changes in external conditions. Here, we address the problem of internally generated variations in pCO$_2$ due to pure carbon-cycle dynamics. We focus on the effect of the strength of Atlantic Meridional Overturning Circulation (AMOC) on such internal variability. Using the Simple Carbon Project Model v1.0 (SCP-M), which we
have extended to represent a suite of nonlinear carbon-cycle feedbacks, we efficiently explore the multi-dimensional parameter space to address the AMOC - pCO$_2$ relationship. We find that climatic boundary conditions, and the representation of biological production in the model are most important for this relationship. When climate sensitivity in our model is increased, we find intrinsic oscillations due to Hopf bifurcations with multi-millennial periods. The mechanism behind these oscillations is clarified and related to the coupling of atmospheric pCO$_2$ and the alkalinity cycle, via the river influx and the sediment outflux.
This mechanism is thought to be relevant for explaining atmospheric pCO$_2$ variability during glacial cycles.

## 1 Introduction

Atmospheric pCO$_2$ values show large variations on many different time scales. Over the Cenozoic, pCO$_2$ values have gradually decreased from values of up to 2,500 ppmv in the Eocene to 300 ppmv at the end of the Pliocene. When considering the Pleistocene glacial-interglacial cycles, one of the remarkable results is the strong correlation between pCO$_2$ and temperature,
with dominant variations of about 100 ppmv in 100,000 years, as reconstructed from ice cores (Petit et al., 1999). Over the industrial period, pCO$_2$ values have increased by 130 ppmv due to human activities (Friedlingstein et al., 2020). This forced trend is superposed on natural variability associated with the seasonal cycle and longer time scale climate variability (Gruber et al., 2019). The effect of the natural variability is much lower than the forced trend on such relatively short time scales. For example, the El Niño- Southern Oscillation (ENSO), a dominant mode of interannual climate variability, induces atmospheric
pCO$_2$ variations of only 1-2 ppmv (Jiang and Yung, 2019). Most studies seek to explain such variations in pCO$_2$ as a forced response of the carbon cycle to changes in external conditions. For example, glacial cycles are thought to be caused by orbital variations in insolation, possibly amplified by physical processes in the climate system (Muller and MacDonald, 2000). Such variations in temperature (and other quantities, e.g., precipitation) then affect the carbon cycle, leading to changes in pCO$_2$. On the other hand, changes in pCO$_2$ will affect global mean temperature and hence may amplify any temperature anomaly. Hence

it is questionable whether the $pCO_2$ response to orbital insolation changes can be considered as a solely forced response, with no internal dynamics of the carbon being involved (Rothman, 2015).

The carbon cycle is comprised of an extremely complex entangled set of processes which act in the different components of the climate system (e.g., land, ocean) on many different time scales. The marine carbon cycle, with its three main carbon pumps is a major player in this cycle, at present-day resulting in the uptake of about 25% of the human released emissions

(Sabine et al., 2004). The carbon pumps involve physical processes, biological processes and processes in ocean sediments. Many carbon cycle feedbacks exist, either between only physical quantities or between biological and physical quantities. An example of such a feedback is the solubility feedback: for higher atmospheric $pCO_2$, solubility of $CO_2$ decreases due to higher ocean temperatures, resulting in relatively less $CO_2$ uptake by the ocean and thus relatively higher atmospheric $pCO_2$. Given this strongly nonlinear system, it would be strange if it would not show strong internal variability, i.e. variability which would

exist even if the carbon-cycle system would be driven by a time-independent external forcing. There are indeed examples (Rothman, 2019), where oscillatory behavior in the carbon cycle has been attributed to internal carbon-cycle dynamics.

The physical context of all carbon pumps is the three-dimensional ocean circulation, which can be roughly decomposed in a wind-driven and an overturning component, the latter strongly related to the deep-ocean circulation. The Atlantic Meridional Overturning Circulation (AMOC) is a major component of the global overturning circulation because of its associated

meridional transport of heat, salt and nutrients.

The relation between the AMOC and atmospheric $pCO_2$ is complicated. A direct effect of a changing AMOC is a change in the distribution of tracers such as temperature, dissolved inorganic carbon (DIC), alkalinity (Alk) and nutrients. For example, after an AMOC weakening the distributions of these tracers affect biological export production via reduced nutrient upwelling (Marchal et al., 1998; Menviel et al., 2008; Mariotti et al., 2012; Nielsen et al., 2019), and gas exchange via changing solubility

of $CO_2$ in the ocean (Menviel et al., 2014). Besides these direct effects, the AMOC also influences mixing in the Southern Ocean. Changes in this mixing due to a weaker AMOC can result in a higher outgoing flux of carbon to the atmosphere (e.g. Schmittner et al., 2007; Huiskamp and Meissner, 2012; Menviel et al., 2014). Furthermore, changes in the AMOC also influence the general sensitivity of the marine carbon cycle to, for example, changes in the wind field (Munday et al., 2014). These processes form a complex puzzle where the sign of atmospheric $pCO_2$ change following an AMOC strength change is

difficult to determine. Currently, different models produce different results with respect to the sign of the atmospheric $pCO_2$ change, which can be attributed to the assessed time scale, model used, and what climatic boundary conditions are used (Gottschalk et al., 2019).

On the other hand, $pCO_2$ also influences the AMOC (Toggweiler and Russell, 2008) and present-day climate models forced with anthropogenic emissions, simulate a weaker AMOC for larger atmospheric $pCO_2$ (Gregory et al., 2005; Weijer et al.,

2020). By contrast, proxy data suggest that in the Last Glacial Maximum both atmospheric $pCO_2$ and the strength of the AMOC were lower (Duplessy et al., 1988). This shows that there is also a sensitivity to climatic boundary conditions in the relation (Zhu et al., 2015) between the AMOC and atmospheric $pCO_2$. The AMOC can also display tipping behavior (Weijer et al., 2019) under an increase of $pCO_2$, which can have large effects on climate. Examples of these effects are disrupted heat transport (Ganachaud and Wunsch, 2000), changing precipitation patterns (Vellinga and Wood, 2002) and a different

distribution of important tracers in the ocean. Such tipping can hence have strong consequences on the carbon cycle, and hence on atmospheric $pCO_2$.

In this paper, we perform a systematic study of internal carbon-cycle variability and the relation AMOC-$pCO_2$ connection, using the Simple Carbon Project Model v1.0 (SCP-M). This model (O'Neill et al., 2019) simulates the most important carbon cycle processes in a simple global ocean box structure. The simple box setup enables us to efficiently scan the parameter space of the carbon-cycle model using parameter continuation methods. With this approach we aim answering the following three questions: (i) How does atmospheric $pCO_2$ respond to changes in the strength of a constant (in time) AMOC? (ii) Does the $pCO_2$-AMOC feedback lead to new variability phenomena? And (iii), are there tipping points and internal oscillations in the carbon cycle?

When answering these questions, we pay special attention to different (non-linear) carbon cycle feedbacks. We will also use two different model configurations to take account of different climatic boundary conditions, the pre-industrial (PI) configuration and the Last Glacial Maximum (LGM) configuration. The SCP-M, its configurations, the different additional feedbacks implemented, and the parameter continuation approach are described in Section 2. In Section 3 we present the results of the different cases considered, and we conclude the paper with a summary and discussion in Section 4.

## 2    Methods

### 2.1    SCP-M

The SCP-M is a carbon cycle box model focused on the marine carbon cycle. Because of its simple structure, it is well suited to test high level concepts in both modern and past configurations. In the ocean several tracers are resolved. In this study we will only use the three most important tracers: DIC, Alk, and phosphate ($PO_4^{3-}$), to reduce the problem size. In the original model there is also a terrestrial biosphere component, and several sources of $CO_2$ to the atmosphere. We will not use these, since our focus is on the marine carbon cycle. The processes which are resolved are ocean overturning circulation, sea-air gas exchange, biological production, calcium carbonate ($CaCO_3$) production and dissolution, and river and sediment fluxes.

The model consists of 8 boxes: 1 atmospheric box and 7 oceanic boxes (Fig. 1). This means that the sediment stock is not explicitly solved for in the model. In the model, the ocean boxes are differentiated on latitude and depth. Consequently, there is no longitudinal variation, and no differentiation between ocean basins. The used boxes are: (1) a low-latitude surface box, (2) a northern high latitude surface box, (3) an intermediate ocean box, (4) a deep ocean box, (5) a southern high latitude surface box, (6) an abyssal ocean box, and (7) a sub-polar surface box. This division in the ocean is based on regions in the ocean where the water masses have similar characteristics. The different boxes are connected via ocean circulation and mixing, which is based upon a conceptual view of the ocean circulation (Talley, 2013). The largest circulation is the Global Overturning Circulation (GOC; $\psi_1$). This circulation connects boxes 4-7 and represents the formation of Antarctic Bottom Water. Next to the GOC, the other major circulation is the AMOC ($\psi_2$) which connects boxes 2-4 and 7. Lastly, there is bidirectional (vertical) mixing between boxes 4 and 6 ($\gamma_1$) and boxes 1 and 3 ($\gamma_2$).

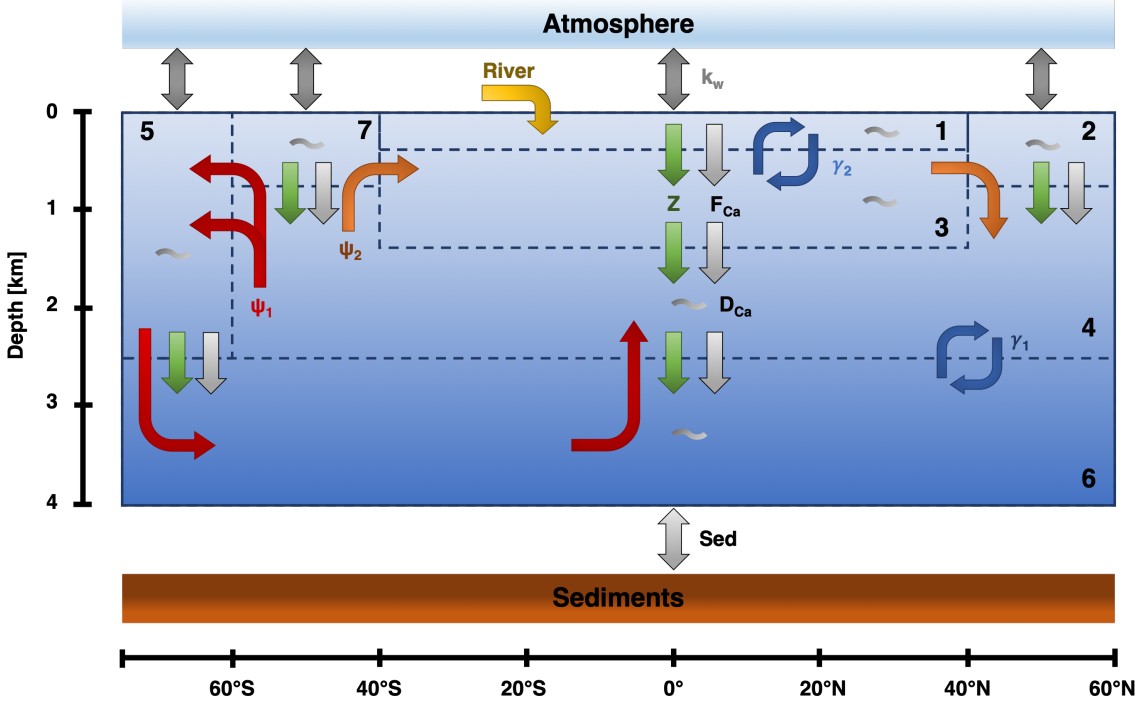

**Figure 1.** The box structure and fluxes for the SCP-M based on O'Neill et al. (2019). $\psi_1$ (red) is the GOC, $\psi_2$ (orange) is the AMOC, $\gamma_1$ and $\gamma_2$ (blue) represent bidirectional mixing. Biological fluxes are represented by the green arrows, calcifier fluxes by the light gray arrows, and general dissolution of calcium carbonate by the grey wiggles in the boxes. $k_w$ (gray) represents the gas exchange between the ocean and the atmosphere. Lastly, there is an influx in Box 1 via the rivers (yellow), and an outflux to the sediments (light gray).

To be able to solve for several fluxes, such as the air-sea gas exchange, the pH in the ocean needs to be determined. Unfortunately, pH is not a conservative tracer, which means that we need a carbonate chemistry module to solve for pH. In the SCP-M, a direct solver is used where the pH value of the previous time step is used as an estimate for the new step (Follows et al., 2006). Using this carbonate chemistry, the model is able to determine the carbonate ($CO_3^{2-}$) concentration and oceanic $pCO_2$. This latter quantity is used to model the exchange of $CO_2$ between the atmosphere and ocean. For each surface box, the flux is proportional to the $pCO_2$ difference between atmosphere and ocean, and a constant piston velocity ($k_w$).

Biological production is constant in the SCP-M. Per surface box, a constant value is used to denote the biological export production at 100 m depth. The organic flux is remineralized in the subsurface boxes following the power law of Martin et al. (1987). The biological export production is also important for the carbonate pump. Via a constant rain ratio, the biological production is linked to the production of calcifiers. Besides organic growth via photosynthesis, calcifiers also take up DIC and Alk to form shells ($CaCO_3$). Upon death, these calcifiers sink to deeper boxes where the shells are dissolved. The dissolution of the shells is dependent on a constant dissolution rate and a saturation dependent dissolution. If the total dissolution of $CaCO_3$

**Table 1.** The parameter values that are different between the two configurations (PI and LGM). In columns 1 and 2 the parameter symbol and description are given. In column 3 the PI value is given and in column 4 the LGM value.

| Parameter | | PI-value | LGM-value |
|---|---|---|---|
| $T_1$ | Temperature box 1 | 23.44 | 17.34 |
| $T_2$ | Temperature box 2 | 9.1 | 3.1 |
| $T_7$ | Temperature box 7 | 5.83 | 0.33 |
| $S_1$ | Salinity box 1 | 35.25 psu | 36.25 psu |
| $S_2$ | Salinity box 2 | 34.27 psu | 35.27 psu |
| $S_5$ | Salinity box 5 | 34.34 psu | 35.34 psu |
| $S_7$ | Salinity box 7 | 34.17 psu | 35.17 psu |
| $\gamma_1$ | Mixing deep - abyssal ocean | 29 Sv | 31 Sv |
| $\psi_1$ | General Overturning Circulation | 29 Sv | 18 Sv |
| $\psi_2$ | AMOC | 19 Sv | 15 Sv |
| $V_n$ | Volume box n | $1 \times V_n$ | $0.97 \times V_n$ |
| $A_n$ | Surface area box n | $1 \times S_n$ | $0.97 \times S_n$ |
| $k_{w_5}$ | Piston velocity box 5 | 3 m/day | 1 m/day |
| $pCO_{2,base}$ | Base atmospheric pCO$_2$ | 244 ppm | 145 ppm |

in the ocean is smaller (larger) than the production in the surface ocean, there is an outflux (influx) of DIC and Alk to (from) the sediments. The river flux for $PO_4^{3-}$ is constant in the SCP-M and balanced by a constant outflux into the sediments. Influx of DIC and Alk via the rivers is variable and related to constant silicate and variable silicate and carbonate weathering. Here the variable component is related to atmospheric pCO$_2$. The difference between the influx of DIC and Alk and the outflux into the sediments determines the change in total carbon and Alk in the system.

A big advantage of the SCP-M is that it has two configurations: a PI configuration, and an LGM configuration. The parameter values in both configurations have been determined via extensive tuning of the model to observations and proxies in O'Neill et al. (2019). The configurations are differentiated on surface ocean temperature and salinity, ocean circulation, sea-ice cover in box 5, and total volume of the ocean. The parameter values of the two different SCP-M configurations can be found in Table 1.

## 2.2 Representation of Carbon-Cycle Processes and Feedbacks

The carbon cycle has many (non-linear) feedbacks which are not represented in the original SCP-M version to keep the model as simple as possible. The absence of these feedbacks can lead to non-physical behavior (e.g. negative concentrations) when parameter values, such as the AMOC strength, are changed. We have implemented several additional feedbacks which can be divided into two categories: those that mostly concern physical processes and those associated with biological processes. The feedbacks are included through parameters $\lambda$'s; when such a parameter is zero, the feedback is not active in the SCP-M and

the original version applies. For all feedbacks, except the feedback on the rain ratio (Eq. 9 below), the sign of the feedback (positive or negative) is unclear beforehand as multiple (carbon cycle) processes are involved.

### 2.2.1 Physical processes

An important feedback is the coupling of temperature to atmospheric $pCO_2$. There are several ways temperature effects the carbon cycle. For example, decreasing temperatures increase the solubility of $CO_2$, which results in more uptake of $CO_2$ by the
ocean. For this feedback, we make a distinction between box 5 and boxes 1, 2 and 7. Box 5, the southern high latitude surface box, is more isolated than the other boxes due to the Antarctic Circumpolar Circulation (ACC). Therefore, we have included the option in the model to use a different sensitivity in Box 5. The temperature in the boxes is calculated as follows

$$T_i = T_{i,base} + \Delta T_i, \quad i = 1, 2, 5, 7 \tag{1}$$

$$\Delta T_i = \lambda_T \times 0.54 \times 5.35 \times \ln \frac{CO_2}{CO_{2_{base}}}, \quad i = 1, 2, 7 \tag{2}$$

$$\Delta T_5 = \lambda_{T5} \times 0.54 \times 5.35 \times \ln \frac{CO_2}{CO_{2_{base}}} \tag{3}$$

Here $T_{i,base}$ is the base temperature in the SCP-M. The change in temperature is dependent on atmospheric $pCO_2$ and a base value of atmospheric $pCO_2$. This base value is the steady state solution in the SCP-M without feedbacks (Table 1). Climate
sensitivity can be changed via the $\lambda$ parameters. For a $\lambda$ of 1, sea surface temperatures increase 2 K per $CO_2$ doubling. As a reference, a 2 K warming for surface air temperatures is at the lower end of the range found in CMIP6 models (Zelinka et al., 2020).

    Besides an effect on solubility, temperature can also affect the piston velocity. In the often used Wanninkhof (1992) formulation, the piston velocity is dependent on temperature via the Schmidt number (equations 4 and 5). In our model, we use this
dependency on the Schmidt number, which causes the piston velocity to increase for warmer temperatures. Hence

$$k_{w,i} = (1 - \lambda_P) \times k_{w,i\,base} + \lambda_P \times k_{w,i\,base} \times (\frac{Sc_i}{660})^{-0.5}, \quad i = 1, 2, 5, 7 \tag{4}$$

Where

$$Sc_i = 2116.8 - 136.25 T_i + 4.7353 T_i^2 - 0.092307 T_i^3 + 0.0007555 T_i^4, \quad i = 1, 2, 5, 7 \tag{5}$$

In these equations, $k_w$ is the used piston velocity, $k_{w,base}$ is the piston velocity in the SCP-M (3 m/day), and T is the temperature
of the box in °C. The $\lambda$ parameter needs to be either 0 (constant piston velocity, as in SCP-M) or 1 (variable piston velocity). Notice that if the temperature feedback is used ($\lambda_T > 0$), the Schmidt number depends on atmospheric $pCO_2$.

### 2.2.2 Biological processes

A large limitation in the original SCP-M is the constant biological production. Nutrient availability introduces a large constraint on biological production but this process is completely absent in the original SCP-M. This process is introduced in the model here by adopting the expression used in the Long-term Ocean-atmosphere-Sediment Carbon cycle Reservoir model (LOSCAR) (Zeebe, 2012). In LOSCAR, production is dependent on the upwelling of nutrients, which in our model translates to the expressions

$$Z_1 = (1 - \lambda_{BI}) \times Z_{1,base} + \lambda_{BI} \times (\gamma_2 \times [PO_4^{3-}]_3 + R_{PO4}) \times \epsilon_1 \tag{6a}$$

$$Z_2 = (1 - \lambda_{BI}) \times Z_{2,base} + \lambda_{BI} \times \psi_2 \times [PO_4^{3-}]_3 \times \epsilon_2 \tag{6b}$$

$$Z_5 = (1 - \lambda_{BI}) \times Z_{5,base} + \lambda_{BI} \times \alpha \times [PO_4^{3-}]_7 \times \epsilon_5 \tag{6c}$$

$$Z_7 = (1 - \lambda_{BI}) \times Z_{7,base} + \lambda_{BI} \times (\alpha \times \psi_1 + \psi_2) \times [PO_4^{3-}]_4 \times \epsilon_7 \tag{6d}$$

In these equations $Z$ represents the production in the surface box, and $Z_{base}$ the value used in the original SCP-M. Furthermore, $\alpha$ is the fraction of $\psi_1$ that moves from Box 4 to Box 7, and $\epsilon$ is the biological efficiency in the box. As with the piston velocity, $\lambda_{BI}$ is either 0 (SCP-M) or 1. Notice that the current branch represented by $\psi_1$ which flows from Box 4 to Box 5, does not influence the production in Box 5. We do not use this branch, since it is assumed to flow into Box 5 below the euphotic zone.

In the equations (6) also the biological efficiency ($\epsilon$) is introduced. There are studies (e.g. Cael et al., 2017) where they relate biological efficiency to temperature. We have adopted a simple linear relation to represent the influence of temperature on biological efficiency, i.e.,

$$\epsilon_i = \lambda_\epsilon \times (-0.1\Delta T_i) + \epsilon_{i,base}, \quad i = 1, 2, 5, 7 \tag{7}$$

In this equation, $\lambda_\epsilon$ controls how strong the relation is between the efficiency and temperature change($\Delta T$). In addition, $\epsilon_{base}$ is the base value of the biological efficiency. These values have been fitted so that $Z$ is equal to $Z_{base}$ under the original parameter values in the SCP-M.

In the SCP-M, $PO_4^{3-}$ is the only nutrient. In the real ocean, additional nutrients play a role in biological production, one of them being nitrate ($NO_3^-$). During photosynthesis, organisms take up nitrate, and thereby increase Alk. This biological influence on Alk is not incorporated in the SCP-M, but present in many other models (e.g. Kwon and Primeau, 2008). We have included this influence as follows:

$$A_{Bio,i} = \lambda_{BA} \times \left(-\frac{16}{106}\right) \times C_{Bio,i}, \quad i = 1, 2, 3, 4, 5, 6, 7 \tag{8}$$

In this equation $A_{Bio}$ is the biological flux affecting Alk. This flux is related to the DIC biological flux ($C_{Bio}$) and the N:C Redfield ratio ($\frac{16}{106}$). For this relation, the $\lambda_{BA}$ parameter can be 0 (not included, original SCP-M), or 1 (included).

Finally, we have also included a feedback for the rain ratio, which is the fraction of calcifiers in the total biological production. In the original SCP-M this is a constant value for all boxes. Calcifiers can be limited in growth when $CO_3^{2-}$ concentrations

are too low. Ridgwell et al. (2007) model this limitation via the saturation state of $CaCO_3$ as

$$F_{Ca,i} = (1 - \lambda_F) \times F_{Ca,base} + \lambda_F \times 0.022(\frac{[Ca]_i[CO_3^{2-}]_i}{K_{sp_i}} - 1)^{0.81}, \quad i = 1, 2, 5, 7 \tag{9}$$

Here, $F_{Ca}$ is the used rain ratio, and $F_{Ca,base}$ is the value used in the original SCP-M (0.07). The saturation state is determined via the concentrations of calcium ($[Ca]$), the carbonate ion concentration $[CO_3^{2-}]$, and an equilibrium constant $K_{sp}$. In this feedback, $\lambda_F$ is either 0 (SCP-M) or 1. The rain ratio feedback is a negative feedback. When carbonate concentrations increase in the surface layer, the rain ratio increases and therefore more calcium carbonate is removed from the surface layer effectively lowering the carbonate concentration.

## 2.3 Parameter continuation methodology

The SCP-M, including our representations of the additional feedbacks, leads to a system of ordinary differential equations of the form

$$\frac{d\mathbf{u}}{dt} = \mathbf{f}(\mathbf{u}(t), \mathbf{p}), \tag{10}$$

where $\mathbf{u}$ is the state vector (containing all the dependent quantities in all boxes), $\mathbf{f}$ contains the right-hand-side of the equations and $\mathbf{p}$ is the parameter vector. Usually, such models are integrated in time from a certain initial condition and the equilibrium behavior is determined for different values of the parameters. However, this is not very efficient to scan the parameter space and, moreover, it is difficult to detect tipping behavior. A much more efficient approach is to determine the equilibrium solutions directly versus parameters, avoiding time-integration, using continuation methods.

Here, we use the continuation and bifurcation software program AUTO to scan the parameter space and detect bifurcations efficiently (Doedel et al., 2007). The SCP-M is very suitable to be implemented into AUTO and to easily compute branches of equilibrium solutions, such as steady states of (10), versus parameters. The equations of the SCP-M turn out to have a singular Jacobian matrix (because both carbon, alkalinity and phosphate quantities are determined up to an additive constant), which requires to add integral conservation equations. We have added such integral conservation equations for carbon (DIC and atmospheric pCO$_2$), Alk and $PO_4^{3-}$ to the model equations to replace the equations for Box 4. The conservation law for $PO_4^{3-}$ is straightforward and already present in the model equations. The constant influx of $PO_4^{3-}$ via the rivers is equal to the constant outflux via the sediments.

In the original SCP-M model, carbon and Alk are conserved in the ocean, atmosphere, continents, and sediments. However, the continental and sediment stocks are not explicitly represented in the version of the SCP-M we use. However, we can describe the change of total carbon and total Alk in the combined atmosphere and ocean stocks over time as

$$\frac{dTC}{dt} = C_{river} \times V_1 + \sum_{n=1}^{7}(C_{carb,n} \times V_n) + \sum_{n=1}^{7}(C_{bio,n} \times V_n) \tag{11a}$$

$$\frac{dTAlk}{dt} = A_{river} \times V_1 + \sum_{n=1}^{7}(A_{carb,n} \times V_n) \tag{11b}$$

In these equations *TC* and *TAlk* are the total carbon and alkalinity in the system. As with $PO_4^{3-}$, total carbon and Alk change due to influx via the rivers ($C_{river}$ and $A_{river}$) and outflux via the sediments. The carbon outflux via the sediments is determined by the sum of carbonate ($C_{carb}$) and biological ($C_{bio}$) fluxes in the system. For Alk, the biological influence is absent. Model simulations with the original SCP-M have shown that the influence of the biological fluxes is negligible, i.e. all biologically produced organic matter is respired in the ocean itself. Therefore, this term can be set to zero in Equation (11a). This makes (11b) proportional to (11a) and hence we include only the latter and use it to determine the change in total Alk in the model.

We also changed the carbonate chemistry in the model. The original SCP-M uses the algorithm of Follows et al. (2006), which solves the carbonate chemistry by using hydrogen ion concentrations from a previous time step. Therefore, the algorithm is inherently transient and, since we directly solve for steady-state solutions, not suitable. We therefore adopted a simple 'textbook' carbonate chemistry based on carbonate alkalinity (Williams and Follows, 2011; Munhoven, 2013). This method approximates oceanic $pCO_2$ by assuming that Alk is equal to carbonate alkalinity ($A_C$=[$HCO_3^-$]+2 [$CO_3^{2-}$]). A disadvantage of this method is that pH values are generally a bit higher (0.15-0.2) than using more complicated algorithms (Munhoven, 2013). These higher pH values are one of the reasons our atmospheric $pCO_2$ values are lower than in the original SCP-M (approximately 60 ppm for case P-CTL described in Section 3).

Eventually, by including (11a) and the overall conservation equations, the version of SCP-M used is a dynamical system with a state vector of dimension $d = 20$. There is one equation for atmospheric $pCO_2$, six for DIC, Alk and $PO_4^{3-}$ in the ocean, and one equation for the total carbon content. Except for the new carbonate chemistry, the necessary changes made to the SCP-M do not change the outcome of the model compared to the original model. When the original model is fitted with the same carbonate chemistry based on carbonate alkalinity, the AUTO implementation and the original code produce the same results.

## 3 Results

In Section 3.1 we present the general sensitivity of atmospheric $pCO_2$ to variations in the AMOC strength. We extend these results in Section 3.2 by adding a coupling between the AMOC strength and atmospheric $pCO_2$. Internal variability found in the model will be presented in Section 3.3. An overview of all cases considered is given in Table 2. Our control experiment uses the original model, which is tuned to accurately represent the pre-industrial and last glacial maximum carbon cycle. From this 'realistic' model we investigate the sensitivity of the carbon cycle to specific carbon cycle feedbacks which can be found in more detailed models. By gradually increasing the amount of feedbacks in the model, we can assess the effects of the (combined) feedbacks.

### 3.1 Sensitivity of atmospheric pCO₂ to the AMOC

In this section the AMOC strength is used as a control parameter and steady states are calculated versus this parameter. For each configuration (PI and LGM) we use three reference cases (x-CTL, the original SCP-M configuration, x-BIO, with a different parameterization for biological production, and, x-ALL, with all feedbacks included, in Table 2, where x is either P for the PI

**Table 2.** Overview of the cases considered and their notation. The left column displays the used feedback. The other columns show the notation and what feedback are included in the specific case. The 'x' in the notation is replaced with either P for the PI configuration, or L for the LGM configuration. Shaded columns indicate that this combination of feedbacks is also used for cases with a coupling between the AMOC and atmospheric pCO$_2$ (Section 3.2). For these cases, 'C' is added to either 'P' or 'L' to denote the coupling. The last column represents the feedback combinations used in Section 3.3. Case x-CTL is the original SCP-M.

| Notation | $\lambda_{BI}$ | $\lambda_T$ | $\lambda_P$ | $\lambda_{BA}$ | $\lambda_F$ | $\lambda_\epsilon$ | $\lambda_{T5}$ |
|---|---|---|---|---|---|---|---|
| x-CTL | 0 | 0 | 0 | 0 | 0 | 0 | 0 |
| x-BIO | 1 | 0 | 0 | 0 | 0 | 0 | 0 |
| x-TEMP | 0 | 1 | 0 | 0 | 0 | 0 | 0 |
| x-PV | 0 | 0 | 1 | 0 | 0 | 0 | 0 |
| x-BALK | 0 | 0 | 0 | 1 | 0 | 0 | 0 |
| x-RAIN | 0 | 0 | 0 | 0 | 1 | 0 | 0 |
| x-BIO-TEMP | 1 | 1 | 0 | 0 | 0 | 0 | 0 |
| x-BIO-PV | 1 | 0 | 1 | 0 | 0 | 0 | 0 |
| x-BIO-BALK | 1 | 0 | 0 | 1 | 0 | 0 | 0 |
| x-BIO-RAIN | 1 | 0 | 0 | 0 | 1 | 0 | 0 |
| x-BIO-TEMP-PV | 1 | 1 | 1 | 0 | 0 | 0 | 0 |
| x-BIO-TEMP(A)-EFF | 1 | 1 | 0 | 0 | 0 | 1 | 1 |
| x-ALL | 1 | 1 | 1 | 1 | 1 | 1 | 1 |
| x-HB | 1 | 2.085 | 1 | 0 | 0 | 1.5 | 1 |

or L for the LGM configuration). The steady-state value of atmospheric pCO$_2$ versus AMOC is shown for the reference cases in Fig. 2, where all branches represent stable fixed points. For the cases where the biological feedback is not included, the solutions for smaller values of AMOC ($< \sim 12$ Sv) display negative PO$_4^{3-}$ concentrations in Box 2 and hence are not allowed. Such boundaries can be automatically monitored in AUTO and the continuation is stopped once a boundary is exceeded.

For the PI-configuration, Fig. 2 shows that, whereas pCO$_2$ increases for larger AMOC strengths in case P-CTL, it remains fairly constant in P-BIO and P-ALL. Atmospheric pCO$_2$ in case P-BIO peaks around 5 Sv, then decreases until approximately 20 Sv after which it increases slightly again. This different behavior occurs because, in case P-BIO, the AMOC has competing influences on DIC concentrations of the surface ocean. A first effect of an increasing AMOC is to increase the ventilation of the deep ocean, which also increases DIC concentrations in the surface layer. This promotes outgassing to the atmosphere. However, by increasing the AMOC strength, biological production in Boxes 2 and 7 is also increased. As a result, DIC and PO$_4^{3-}$ are transported from the surface layer to the deep ocean. The first effect is dominant after 20 Sv, and the second effect in the range of 5 to 20 Sv. The absence of the second effect in P-CTL explains the difference in sensitivity between P-CTL and P-BIO. P-ALL behaves fairly similar as P-BIO, except in the regime with a weak AMOC strength ($< \sim 4$ Sv). This behavior is caused by the saturation dependent rain ratio. When we look at the other cases (Fig. 3a), we see that they either behave qualitatively like P-CTL (the cases without 'BIO'), or P-BIO (cases with 'BIO'). Looking in more detail, we can see that

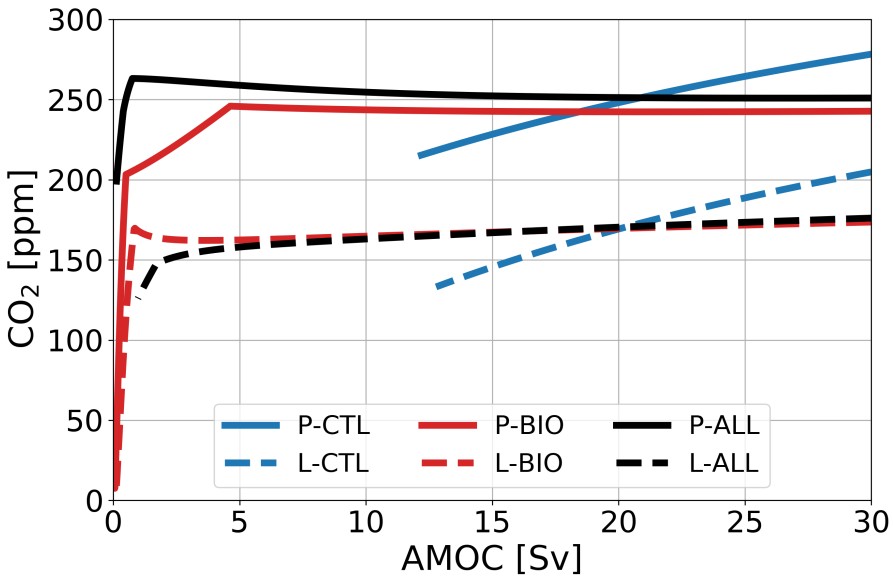

**Figure 2.** Atmospheric pCO₂ in ppm under varying AMOC strength in Sv for three reference cases (blue: no feedback, red: with biological feedback, black: all feedbacks) in two configurations (solid: PI, dashed: LGM). All branches represent stable fixed points.

(a)

(b)

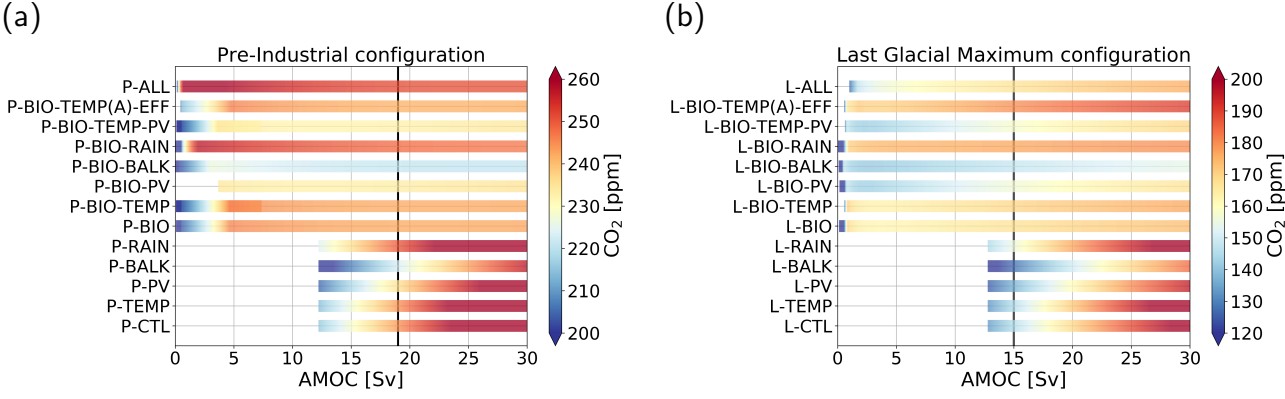

**Figure 3.** Atmospheric pCO₂ in ppm (color shading) under varying AMOC strength in Sv for the cases considered in Table 2. (a) Pre-industrial configuration. (b) Last Glacial Maximum configuration. Note that the range of the color shading differs between the two configurations and that some CO₂ concentrations fall outside the displayed range. The AMOC range of the bars differ, because for some cases the steady solution becomes nonphysical (e.g. negative concentrations or large subzero temperature). The vertical black lines represent the AMOC strength in the original SPC-M.

when we include the rain ratio feedback (cases P-RAIN, P-BIO-RAIN) atmospheric $pCO_2$ is higher, and when we include the biological influence on alkalinity, atmospheric $pCO_2$ is lower (cases P-BALK, P-BIO-BALK). The results in Fig. 3a show that the biological feedback ($\lambda_{BI}$=1) is the most dominant feedback in the PI configuration, i.e., including this feedback leads to a completely different sensitivity of the carbon cycle to changes in the AMOC strength.

For the LGM configuration (Fig. 2), two important differences with respect to the PI-configuration appear: (1) atmospheric $pCO_2$ is approximately 80 ppm lower, and (2) cases L-BIO and L-ALL have a different sensitivity than cases P-BIO and P-ALL for lower AMOC values. Where in P-BIO atmospheric $pCO_2$ decreases for an increasing AMOC between 5 and 20 Sv, L-BIO shows a monotonous increase of atmospheric $pCO_2$ from 3 Sv onward. We see this different relation, because in the LGM-configuration, deep-ocean ventilation, which can be seen as the sum of the GOC and AMOC, is lower due to a weaker GOC. Consequently, deep-ocean ventilation is more sensitive to changes in the AMOC. This eventually causes the different response of cases L-BIO and L-ALL with respect to case P-BIO and P-ALL. Cases L-TEMP to L-BIO-TEMP(A)-EFF (Fig. 3b) relate to the L-CTL and L-BIO as in the PI-configuration. Fig. 3b shows that in the LGM configuration, as is the case in the PI configuration, the biological feedback is most dominant. The other feedbacks only influence the offset of $CO_2$ concentrations, but do not result in large changes to the relation between the AMOC and atmospheric $pCO_2$.

## 3.2 Coupling AMOC - carbon cycle

The AMOC strength depends also on atmospheric $pCO_2$ and below we will discuss the steady state model solutions when a coupling between the AMOC and atmospheric $pCO_2$ is applied. This coupling is based on how the AMOC responds to increasing atmospheric $pCO_2$ in CMIP6 models (e.g. Bakker et al., 2016) and given by

$$\psi_2 = \psi_{2,base} \times (1 - \lambda_A \times 0.1 \times 0.54 \times 5.35 \times \ln \frac{CO_2}{CO_{2_{base}}}) \tag{12}$$

In this equation $\psi_{2,base}$ is a base value of the AMOC taken from the uncoupled case (where the AMOC is prescribed), $\psi_2$ is the actual AMOC strength in $m^3$/s in the coupled case and $\lambda_A$ determines the strength of the coupling. We use three different values of $\lambda_A$ in this section: (1) 0 (no coupling), 1 (20 % decrease for a $CO_2$ doubling), and 4 (80% decrease for a $CO_2$ doubling). As the AMOC strength $\psi_2$ is now part of the state vector, we need other quantities as control parameters. We will use three different parameters here: (1) the rain ratio ($F_{Ca}$), (2) the biological production ($Z$), and (3) the piston velocity ($k_w$). We have chosen these three parameters since they (approximately) represent the three carbon pumps: the carbonate pump, the soft tissue pump, and the solubility pump, respectively. We follow the steady-state solution in these parameters, where possible, between 0.1 to 10 times the reference value (indicated by the multiplier in Fig. 4). This large, though not necessarily realistic, range is used to test the sensitivity of the carbon cycle to the parameters, and to see whether bifurcations can arise in the carbon cycle. When we look at the effect of increasing $\lambda_A$, i.e. the coupling, we see that the general sensitivity of the solution to changes in model parameters decreases. This effect is best seen in case LC-CTL, but also present in the other cases, though less pronounced.

In Fig. 4a, b we plot the results when we use the rain ratio as a control parameter in the continuation. There are no large differences between the different cases and configurations. Generally, we see two regimes. For low rain ratios, the solution is

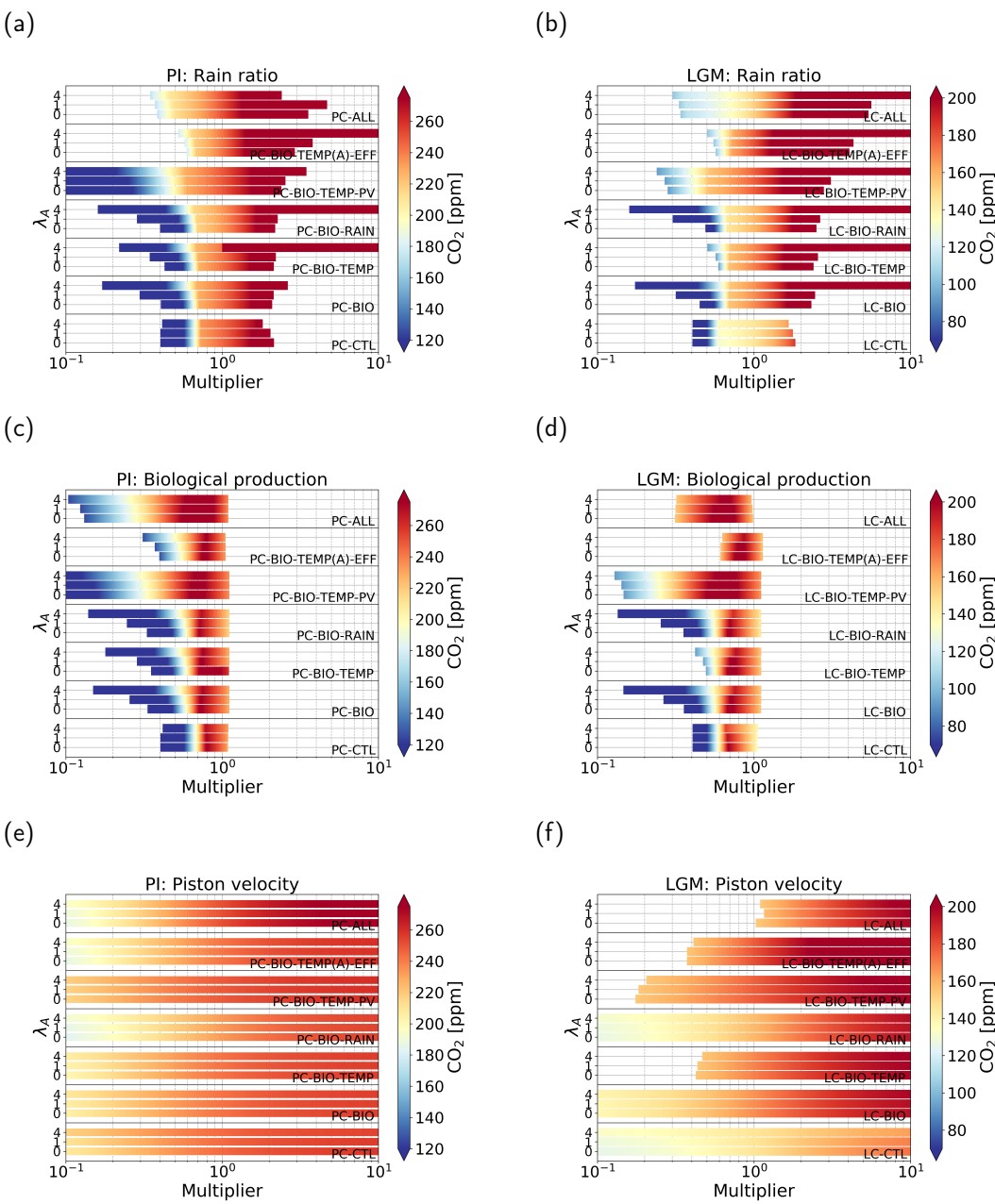

**Figure 4.** Atmospheric $pCO_2$ in ppm (color shading) under varying parameter values. The left column represents cases of the PI configuration, and the right column of the LGM configuration. The top row shows cases where the strength of the rain ratio is varied between 0.1 and 10 times the original value. The other rows show the same but for cases where biological production (middle row) and the piston velocity (bottom row) are varied. In total 7 feedback combinations are used, denoted by the text within the graph. For each case, three different coupling strengths have been used: (1) $\lambda_A$=0, (2) $\lambda_A$=1, and (3) $\lambda_A$=4.

quite sensitive to changes in the rain ratio. Where the coloring in Fig. 4a, b is yellow (around 230 ppm for the PI and around 140 ppm for the LGM configuration), we see a shift: the solution becomes less sensitive to changes in the rain ratio. To explain the regimes of sensitivity, we note that the $CaCO_3$ production is linearly related to the rain ratio. The production minus the dissolution of $CaCO_3$ in the water column determines the outflux of Alk and DIC via the sediments. The different regimes can

be explained by the amount of $CaCO_3$ dissolution in the deep and abyssal ocean. For low rain ratios, the saturation state in the ocean is larger than 1, which means there is no saturation driven dissolution and only constant dissolution. This makes the outflux of Alk and DIC linearly proportional to the production: if the rain ratio is low, the outflux is also low. Because we are looking at a steady state solution, this decrease in burial has to be compensated for by a weaker influx, i.e. a lower river influx. This is only possible when atmospheric $pCO_2$ is lower. For larger rain ratios, we have both saturation dependent and constant

dissolution in the subsurface boxes, i.e. more dissolution in the water column. Due to the variable dissolution, the outflux of Alk and DIC is no longer fully determined by $CaCO_3$ production. This results in a lower sensitivity of the outflux to changes in the rain ratio. Therefore, atmospheric $pCO_2$ is also less sensitive to the rain ratio.

For biological production as a control parameter (Fig. 4c, d) again all cases show comparable behavior. We can see that the parameter range for higher biological production is short. This is because $PO_4^{3-}$ concentrations become negative at this

point, even when we include the biological feedback. All cases have a maximum in atmospheric $pCO_2$ around 0.7-0.8 times the original value. When the multiplier is lower than this value, we see a positive relation (higher biological production, higher atmospheric $pCO_2$). For values larger than the maximum, we see an opposite relation, i.e. lower atmospheric $pCO_2$ for higher biological production. This second regime is generally what we would expect when biological production is increased, i.e. when biological production removes more carbon from the surface layer, more carbon can be taken out of the atmosphere

by the surface ocean which reduces atmospheric $pCO_2$. The first regime is not what we would expect at first, but this can be explained by the same mechanism as for the rain ratio: reduced biological production leads to low production of $CaCO_3$ leading to low burial rates of $CaCO_3$. Lower burial rates should lead to lower river influx because the sources and sinks of alkalinity to the ocean should balance, which can only be achieved by decreasing atmospheric $pCO_2$. Again, increasing the AMOC coupling only reduces the sensitivity of the solutions.

In Fig. 4e-f, we plot the results when we use the piston velocity parameter ($k_w$) as a control parameter. By the gradually changing colors, we can see a logarithmic relation with higher sensitivities for lower piston velocities. The different feedbacks, configurations, and coupling strengths have the same effect as for the other two control parameters discussed above.

### 3.3 Internal oscillation

The feedback strengths we have used so far have been quite modest. The continuation methodology enables us to efficiently

look at cases with different feedback strengths and to see whether different combinations can induce bifurcations in the carbon cycle and by extensively scanning the parameter space we found interesting results. Especially in the LGM-configuration, when climate sensitivity ($\lambda_T$) and the biological efficiency feedback ($\lambda_\epsilon$) are increased, bifurcations arise on the branches of steady solutions. With case L-HB (for parameter values, see Table 2), we present an example where we find a supercritical Hopf Bifurcation (HB) around 13 Sv (Fig. 5a) in the uncoupled case ($\lambda_A = 0$, so the AMOC strength is a control parameter

again). The HB produces a stable limit cycle extending to larger AMOC strengths with a period between 5,000 and 6,000 years where all state variables oscillate. In this section we look at the internal oscillation at 15 Sv (Fig. 5b). The oscillation has a period of 5,814 years, and atmospheric $pCO_2$ has a range of 72 ppm.

The HB described in this section exists for a large range of parameter values and is thus robust. One important constraint on the existence of the bifurcation is the coupling strength between atmospheric $pCO_2$ and biological production. This coupling

comes down to the effect of atmospheric $pCO_2$ on the biological efficiency ($\epsilon$), which can be increased by increasing the temperature feedback ($\lambda_T$) and/or the efficiency feedback ($\lambda_\epsilon$). We do not find this bifurcation in the PI-configuration, because when the biology feedback ($\lambda_{BI}$=1) is included, atmospheric $pCO_2$ is insensitive to changes in the AMOC strength (case P-BIO, Fig. 2). Because of this low sensitivity, surface ocean temperature and biological efficiency are also insensitive to changes in the AMOC strength in the PI-configuration. Therefore, the coupling between the two is less effective in this configuration and we do not find a HB.

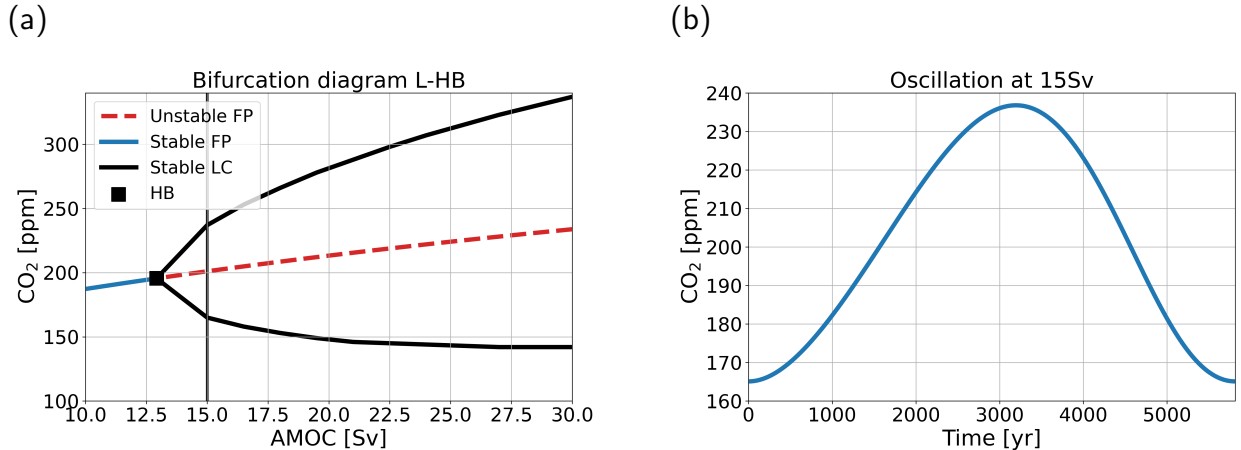

**Figure 5.** (a) Bifurcation diagram for case L-HB in atmospheric $pCO_2$ - AMOC space. Blue, solid lines denote stable steady states (or fixed points, FPs); red, dashed lines indicate unstable steady states; black, solid lines indicate a stable limit cycle (LC), and the black square denotes the location of a (supercritical) Hopf Bifurcation (HB). (b) The oscillation of atmospheric $pCO_2$ in ppm versus time in years for the limit cycle at 15 Sv. The period is 5,814 yr.


To explain the mechanism behind the oscillation, we have to look at the time-dependent solution of the model. What is important for this oscillation is the coupling between atmospheric $pCO_2$ and the alkalinity cycle. Alkalinity influences the gas exchange between the ocean and the atmosphere via the carbonate chemistry and is, in turn, influenced by atmospheric $pCO_2$ because the source and sink of alkalinity are coupled to $pCO_2$. The source, the river influx, is directly proportional to

atmospheric $pCO_2$ (Eq. 13).

$$C_{river} = W_{SC} + (W_{SV} + W_{CV}) \times pCO_2^{atm} \qquad (13)$$

Where $W_{SC}$ is a parameter reflecting constant silicate wheatering, $W_{SV}$ a parameter representing variable silicate weathering, and $W_{CV}$ a parameter representing variable carbonate weathering. $pCO_2^{atm}$ represents the partial pressure of $CO_2$ in the atmosphere. This relation was already used in the SCP-M and is directly taken from Toggweiler and Russell (2008), meaning no adaptations were made to the relation, nor the parameter values. This expression is adapted from a formulation used in Walker and Kasting (1992).

The sink, i.e. outfluxing via the sediments, is related to $CaCO_3$ burial, which is the difference between $CaCO_3$ production in the surface ocean and $CaCO_3$ dissolution in the ocean and sediments. As discussed in Section 3.2, $CaCO_3$ dissolution consists of two parts: a saturation driven part and a constant part (Eq. 14).

$$C_{diss} = ([CO_3^{2-}][Ca^{2+}]) \times k_{Ca} \times (1 - (\min(( \frac{[CO_3^{2-}][Ca^{2+}]}{K_{sp}} ), 1)))) + D_C \tag{14}$$

Where the first part is related to the saturation state: $\frac{[CO_3^{2-}][Ca^{2+}]}{K_{sp}}$. If the saturation state is larger than 1, the saturation dependent dissolution is 0, and only the constant term remains ($D_C$). In the oscillation, the saturation state of $CaCO_3$ in the ocean is everywhere larger than 1. This happens when the river influx is larger than the biogenic flux in the surface ocean (Zeebe and Westbroek, 2003), which is a plausible for the past ocean. Therefore, total dissolution in the ocean is constant and does not vary. This means that $CaCO_3$ burial becomes a function of $CaCO_3$ formation and thus biological production. Since this production is dependent on the biological efficiency, which is directly proportionate to atmospheric $pCO_2$, the sink is also influenced by atmospheric $pCO_2$. However, the effect of atmospheric $pCO_2$ on the source and sink is opposite. When atmospheric $pCO_2$ is high, the river influx is high, while the sediment outflux is low. This is key to the general mechanism sketched in Fig. 6.

The results show that atmospheric $pCO_2$ is affected by the amount of ingassing into Box 1. Therefore, we start the explanation of the oscillation in Fig. 6 at this point. At the beginning of the oscillation (time t = 0 in Fig. 6), ingassing in Box 1 starts to decrease. As a result, atmospheric $pCO_2$ starts to increase approximately 200 years later. There is a delay, since atmospheric $pCO_2$ is not solely determined by the gas exchange with Box 1. The increase in atmospheric $pCO_2$ has multiple effects. First of all, temperatures start to increase, which lowers biological efficiency. This in turn reduces $CaCO_3$ production, and thus the sink of alkalinity is also reduced. Another effect of increasing atmospheric $pCO_2$, is an increasing river flux, i.e. an increasing source of alkalinity into the ocean. After a quarter period (time t = T/4 in Fig. 6), the source becomes larger than the sink, and total alkalinity in the ocean starts to increase. Meanwhile, atmospheric $pCO_2$ is still increasing. As a result, the river influx also keeps increasing, while the sediment outflux keeps decreasing. After half a period (time t = T/2 in Fig. 6), oceanic $pCO_2$ in Box 1 starts to decrease because alkalinity concentrations in Box 1 have increased. The lower oceanic $pCO_2$ causes ingassing into Box 1 to increase, which in turn decreases atmospheric $pCO_2$. The other half of the period is as explained above, but then the opposite.

The processes described above are important for driving the oscillation. However, these are not the only processes resolved in the model. The concentrations of DIC, Alk and $PO_4^{3-}$ in the ocean boxes are subtle balances of multiple larger fluxes where the sum of these fluxes can be more than 100 times smaller than the individual fluxes. It is therefore difficult to assess the

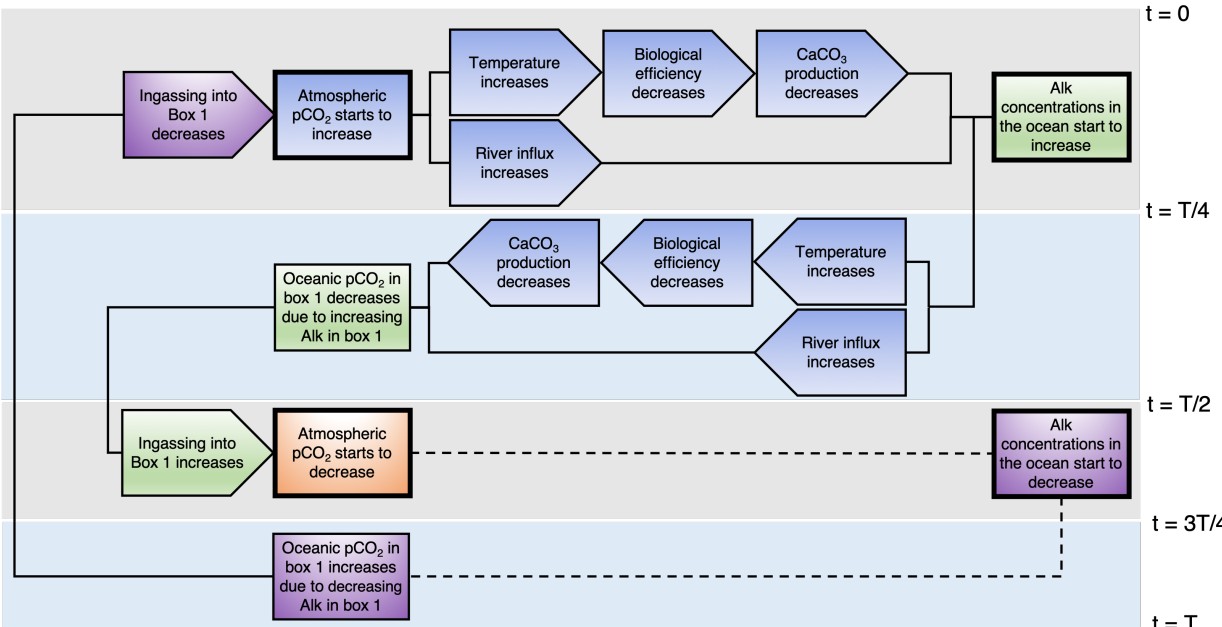

**Figure 6.** Schematic representation of the mechanism of the internal oscillation. The rectangles represent state variables, while the pointed blocks represent fluxes or model parameters. Some boxes have thicker outlining. These boxes cause a chain of events. The chain corresponding to the box, are the boxes with the same color shading. The gray and blue rectangles in the background represent a quarter period. In the second half of the period, processes are replaced with a dashed line. These processes are the opposite of what happens in the first half of the period.

effects of all the individual fluxes, since they also depend on each other in this system. We do see that the DIC concentrations in the surface ocean boxes lag atmospheric $pCO_2$ by multiple centuries. It thereby increases oceanic $pCO_2$ after atmospheric $pCO_2$ has reached it maximum, which dampens the amplitude of the oscillation. The solubility constant ($K_0$) and dissociation constants ($K_1$ and $K_2$), which are also important for the air-sea gas exchange, oscillate due to the dependency on temperature and also dampen the amplitude of the oscillation. It is good to note that all these processes are responsible for the exact shape

and amplitude of the oscillation. However, the coupling between atmospheric $pCO_2$ and the alkalinity cycle appears to be the driving mechanism.

     In Fig. 7a, we can see that total alkalinity in the ocean lags atmospheric $pCO_2$ by approximately a quarter period. In Fig. 7b we can also see the anti-correlation between the source and sink of alkalinity to the ocean. Comparing the sink and source, we can clearly see a strong (anti-) correlation between atmospheric $pCO_2$ and the (sink) source of alkalinity. The anti-correlation

between the source and sink is the driving mechanism behind the oscillatory behavior. It is good to point out that the amplitude of the sink of alkalinity is larger than that of the source. The time scale of the oscillation ($\sim$ 6,000 years) is related to the adjustment time of $CO_3^{2-}$ to an imbalance between the influx and outflux of alkalinity and DIC in the ocean. This process, termed the calcium carbonate homeostat (Sarmiento and Gruber, 2006), has a timescale between 5 and 10 kyr (Archer et al.,

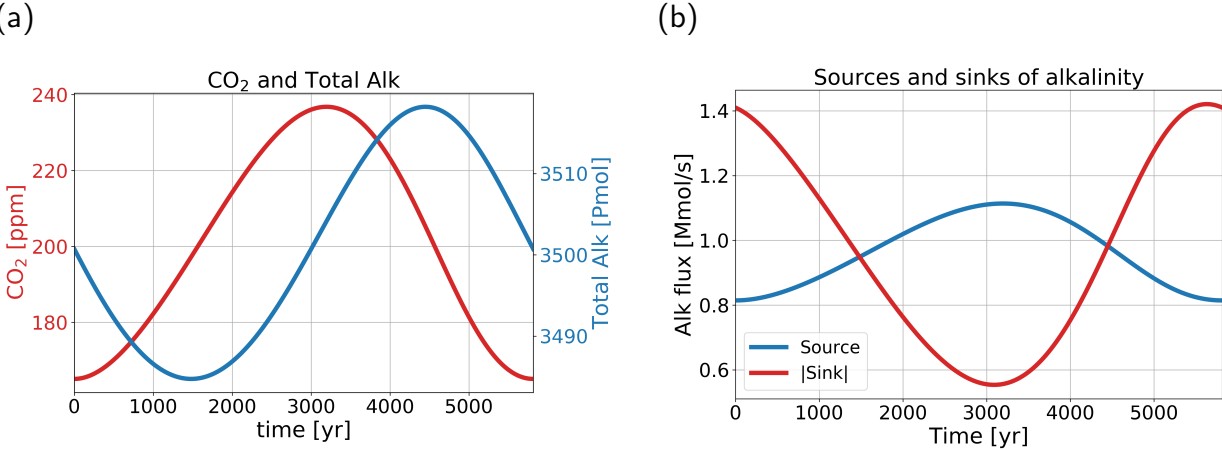

**Figure 7.** (a) Atmospheric $pCO_2$ (red, left y-axis) in ppm, and total alkalinity (blue, right y-axis) in the ocean in Pmol for one oscillation. Total alkalinity lags $pCO_2$ by approximately a quarter period. (b) The source (blue) and the absolute value of the sink (red) of alkalinity in the ocean. The source represents river influx, and the sink represents the sediment outflux. When the lines cross, i.e. around 1,500 yr and 4,400 years, total alkalinity in (a) has a minimum and a maximum respectively.

1997). The period of the internal oscillation corresponds well to this range. The river influx, which plays a role in the oscillation, is usually viewed as a slow process with time scales of the order of 10 kyr or more. In the oscillation however, the river flux seems to be active on shorter time scales. This is because the system does not reach equilibrium and continuously oscillates. The river flux responds directly to the oscillations in atmospheric $pCO_2$, which are influenced by processes on time scales shorter than that of the river flux. It is also good to note that, even though it seems box 1 is a main driver in the oscillation, it is in fact a global process due to the role of $CaCO_3$ burial.

## 4 Summary and discussion

In this study we investigated steady states of an extended version of the simple carbon cycle box model (SCP-M), where additional feedbacks have been included. Focus was on the relation between the AMOC and atmospheric $pCO_2$ for these steady states, with a special attention to the effect of feedbacks and climatic boundary conditions on this relation. Although the model we use is a simple box model, the original SCP-M was shown to be quite capable of simulating present-day observations and proxy data (LGM) (O'Neill et al., 2019).

In Section 3.1 we looked into how the carbon cycle, and specifically atmospheric $pCO_2$ responds to changes in the AMOC. These cases include different combinations of additional feedbacks. Our results (Section 3.1) suggest that the most important feedback, is the biological feedback, represented by equation (6). In both the PI and the LGM configurations, this feedback leads to a different sensitivity of atmospheric $pCO_2$ to the AMOC (Fig. 2). Other feedbacks did not introduce large effects on the sensitivity (Fig. 3). This shows that biology can exert a large effect on atmospheric $pCO_2$, which support studies with

more detailed models where biological production plays a role in the response of atmospheric $pCO_2$ to changes in the AMOC (e.g. Nielsen et al., 2019). The results also show the importance of the climatic boundary conditions, as was already stated in Gottschalk et al. (2019). Generally, cases with the biological feedback (x-BIO, and other cases including 'BIO') respond differently in the LGM configuration than in the PI configuration. This is related to the difference in deep ocean ventilation between the two configurations.

When a coupling between the AMOC and atmospheric $pCO_2$ is included (Section 3.2), the $pCO_2$ of the steady solutions becomes less sensitive to changes in model parameters ($k_w$, $Z$, $F_{Ca}$). This shows that the coupling works as a negative feedback in the carbon-cycle dynamics. What is interesting to see, is that the carbon cycle feedbacks do not have a large effect on the AMOC- $pCO_2$ relation. This implies that ocean circulation is very effective in damping changes in gas exchange ($k_w$), biological ($Z$) and $CaCO_3$ ($F_{Ca}$) production.

When considering bifurcations of the steady solutions, an important result is what we did not find: saddle-node bifurcations. Hence, although quite nonlinear carbon cycle processes have been captured in this model, no multiple equilibrium regimes and associated hysteresis occur. As a consequence, any sharp transition in carbon cycle quantities cannot be easily linked to a transition between different steady states. However, we did find internal oscillations in the model, in particular with a period of 5,000 to 6,000 years related to the $CaCO_3$ homeostat (Fig. 6). Important for this oscillation is the process representation that $CaCO_3$ production reduces for increasing temperatures, which is supported by studies that suggest a decreased production under high atmospheric $CO_2$ concentrations (Barker and Elderfield, 2002). However, this assumption is under debate as there are studies that find an increased calcifier production for higher temperatures (Cole et al., 2018) in specific situations. Whether this internal oscillation also exists in a system where the AMOC strength and atmospheric $pCO_2$ are coupled (as in Section 3.2), is uncertain. The internal oscillations was found using the AMOC strength as control parameter, which is not possible with a relation as in Section 3.2. The results in Section 3.2 show that the system is rather insensitive to this feedback, making it possible that the oscillation also exists in this coupled system.

Linking this oscillation to proxy data is difficult, especially since the variation in atmospheric $pCO_2$ is relatively high (72 ppm) for reasonable AMOC values. If we look for example at the record of the last glacial period, $pCO_2$ variations are of the order of 20 ppm (Bauska et al., 2021). The variation found in our model is closer to that during the Pleistocene glacial cycles, but on a much shorter time scale. The time scale is actually closer to that of the Heinrich events. It is therefore hard to find an oscillation like this in the past record, but this does not mean the mechanism is not relevant. If we look at more fundamental work, our mechanism shares similarities to the internal oscillation found in a conceptual model where only Alk and DIC are resolved (Rothman, 2019). The mechanism in Rothman (2019) is based on the imbalance between the influx and outflux of DIC in the surface ocean, and is thus comparable to our mechanism. The phase differences in our model between quantities in the carbonate system (i.e. DIC, Alk, pH, $CO_3^{2-}$, $HCO_3^-$, and $H_2CO_3$) in the top 250m compare well to those in Rothman (2019) (not shown). However, the responsible processes are different. In Rothman (2019) there is an important role for respiration of organic matter. In our model, this flux is implicitly modeled and we can reconstruct a similar flux from the export production. This reconstructed flux has comparable phase differences with the carbonate content as in Rothman (2019), but the relative strength of the flux does not match the burial flux in our model. This means that the SCP-M captures a different

internal oscillation. In Rothman (2019) there is an important role for the ballast feedback because it couples the sources and sinks of DIC using the carbonate-ion concentration. In our oscillation, it is not the ballast feedback that drives the oscillation, but the $CaCO_3$ homeostat, coupling the sources and sinks of alkalinity through atmospheric $pCO_2$.

The results in this study are achieved with a very simple framework with multiple assumptions and limitations. The main assumption we make is that the SCP-M is a well performing model for the Last Glacial Maximum and Pre-industrial periods. Comparison of the model data with observations in O'Neill et al. (2019) support this assumption and show that the SCP-M performs well in both time periods. In the model, the river influx, a process important for the oscillation, is also subject to assumptions. Assumptions in the river flux parameterization that possibly affect the oscillation are the parameter values and the fact there is no delay between the river influx and atmospheric $pCO_2$. The parameter values are important for the amplitude of the oscillation and decreasing the parameters would result in a decrease in amplitude of the oscillation. However, the assumed parameters are fitted to represent estimated river influx in present day conditions, which suggests that the used parameter values are probable. The timing, i.e. the fact that there is no time delay between atmospheric $pCO_2$ and the river influx is to some extent important. The approximate quarter period delay between atmospheric $pCO_2$ and total alkalinity (and alkalinity in box 1) is important for driving the oscillation. The river influx plays a role in this by changing the alkalinity in box 1. Unless the time delay is increased up to multiple centuries, we do not expect qualitatively different results. Furthermore, we expect this relation to be valid since the time scales and the model complexity are similar to the origin of this formulation (Walker and Kasting, 1992). Besides some assumptions in the original model, we also made some assumptions for the features we added to the model. The first assumption we made here is that the changes to the model do not make refitting of the parameters necessary. For most changes we made, we do expect this assumption to be valid since for most features the elemental cycles remained the same and constant parameter values were replaced by equations which keep the parameter values close to their original values. The addition of the biological alkalinity flux might make refitting of parameters necessary since a complete new process is added to the alkalinity cycle. Refitting of the parameters would be a large exercise and would also make comparison between the different cases difficult. This means that cases where this feedback is used should be approached more carefully. However, cases with this feedback do not show divergent results compared to other cases. Maybe the most impactful change we made is the simplification of the carbonate chemistry. This change typically reduces pH by 0.15-0.2 (Munhoven, 2013), makes atmospheric $pCO_2$ more sensitive to changes, and change equilibrium $pCO_2$ values by 20% (Munhoven, 2013) explaining the approximate 60 ppm lower atmospheric $pCO_2$ in our model. We do not expect general sensitivities as discussed in our results to change a lot, though the magnitude of atmospheric $pCO_2$ values need to be viewed critically. The assumption that biological efficiency is linearly related to change in temperature might not be valid while this assumption is important for the driving mechanism of the internal oscillation. However, what seems to be important for the oscillation is that the coupling between atmospheric $pCO_2$ and the biological efficiency is strong enough and not necessarily the exact formulation of the feedback. Limitations of the model include the incapability to discern between ocean boxes and strict, slightly arbitrary box boundaries. Due to these limitations, this model is not suitable to look at local processes. However, the original SCP-M simulates representative global values, making it suitable for the application in this study.

In this study we have scanned large ranges of parameter values, with some values outside realistic ranges. The parameters we have varied are AMOC strength, the rain ratio, biological production, the piston velocity and climate sensitivity. It is good to note that the wide range we have used is not necessarily realistic, but still yields valuable information. By using such a wide range for certain parameters, we can be quite certain that there are no saddle node bifurcations and therefore no multiple equilibria in realistic parameter ranges. We believe that most results are within a realistic range for the AMOC

strength since present day model simulations show maximum AMOC strengths of around 25 Sv (Weijer et al., 2019), while model simulations simulating an AMOC collapse show very weak AMOC strengths. In Section 3.2, we studied a large range of rain ratio, biological production and piston velocity values. The main purpose of this large range was to see whether bifurcations would arise, which did not occur. Most of the ranges studied here are unrealistic, however, sensitivities of the model to changes in the carbon pumps are shown by the results yielding information about the behavior of this model. The climate sensitivity

variations we used are all within CMIP6 ranges (1.8K-5.6K; Zelinka et al., 2020). Runs without the temperature feedback however, may also yield unrealistic results since ocean temperatures remain above freezing temperature even for near zero atmospheric $pCO_2$ values. Therefore, cases without the temperature feedback can be deemed unrealistic for low atmospheric $pCO_2$.

    In conclusion, we have found that the relation between atmospheric $pCO_2$ and the AMOC strength relies mostly on biological

processes and climatic boundary conditions. Therefore, we suggest that by comparing results of different models, special attention should be given to the way biological production is represented. Our study also shows that atmospheric $pCO_2$ appears to be rather insensitive to changes in the AMOC strength, which suggests that projected weakening of the AMOC in the future does not lead to a large response in atmospheric $pCO_2$. In this study we also searched for saddle-node bifurcations, but we did not find any, suggesting that tipping points in the carbon cycle are unlikely to occur. Our most interesting result is the discovery

of an internal oscillation in the carbon cycle and we hope that the mechanism behind this oscillation will stimulate further model work and be useful for explaining past atmospheric $pCO_2$ variability.

*Code and data availability.*    The original version of the SPC-M is available at https://doi.org/10.5281/zenodo.1310161, the AUTO implementation is available at https://github.com/dboot0016/SCPM-AUTO (Boot et al., 2021). AUTO can be downloaded from https://sourceforge.net/projects/auto-07p/.

## Appendix A: Model parameters

In this appendix values and descriptions of the parameters in the extended SCP-M are given. In Tables A1 to A3 the parameter values used in our model are presented. The values presented here are for the pre-industrial configuration. The parameter values that are different in the Last Glacial Maximum configuration are presented in Table 1. All parameter values, except the biological efficiency ($\epsilon$) parameters, are taken from the SCP-M. The biological efficiency parameters have been fitted such that

$Z$ in equation 6 is equal to $Z_{base}$ under the original parameter values in the SCP-M. When determining the value of $\epsilon_{2,base}$,

we also took the effect of the biological production in the North Pacific into account, which leads to a value of $\epsilon_{2,base} > 1$. In Table A4 we also present the literature where the expressions for the equilibrium constants were taken from.

**Table A1.** Symbol (column 1), description (column 2), value (column 3), and units (column 4) of the general parameters used in our model.

| Symbol | Description | Value | Units |
|---|---|---|---|
| $V_{at}$ | Volume of the atmosphere | $1.76 \times 10^{20}$ | m$^3$ |
| $\rho$ | Sea water density | 1029 | kg m$^{-3}$ |
| $F_{Ca,base}$ | Base rain ratio | 0.07 | - |
| $n$ | Order of CaCO$_3$ dissolution kinetics | 1 | - |
| $P_C$ | Mass percentage of C in CaCO$_3$ | 0.12 | - |
| $D_{Ca}$ | Constant dissolution rate of CaCO$_3$ | $2.75 \times 10^{-13}$ | mol m$^{-3}$ s$^{-1}$ |
| $W_{SC}$ | Constant silicate weathering | $2.4 \times 10^{-12}$ | mol m$^{-3}$ s$^{-1}$ |
| $W_{SV}$ | Variable silicate weathering parameter | $1.6 \times 10^{-8}$ | mol m$^{-3}$ atm$^{-1}$ s$^{-1}$ |
| $W_{CV}$ | Variable carbonate weathering parameter | $6.3 \times 10^{-8}$ | mol m$^{-3}$ atm$^{-1}$ s$^{-1}$ |
| $k_{CaCO3}$ | Constant CaCO$_3$ dissolution rate | $4.4 \times 10^{-6}$ | s$^{-1}$ |
| $R_{PO4}$ | River influx of PO$_4^{3-}$ | $1.5 \times 10^4$ | |
| $b$ | Exponent in Martin's law | 0.75 | - |
| $d_0$ | Reference depth for biological productivity | 100 | m |
| $\alpha$ | Fraction of the GOC that flows through Box 7 | 0.5 | - |
| $\gamma_1$ | Bidirectional mixing between Box 4 and 6 | 29 | Sv |
| $\gamma_2$ | Bidirectional mixing between Box 1 and 3 | 40 | Sv |
| $\psi_1$ | General overturning circulation | 29 | Sv |
| $\psi_{2,base}$ | Base value of the Atlantic Meridional Overturning Circulation | 19 | Sv |
| $k_{w,base}$ | Base piston velocity | 3 | m/day |
| $R_{C:P}$ | Redfield C:P ratio | 130 | mol C/mol P |
| $R_{P:C}$ | Redfield P:C ratio | 1/130 | mol P/mol C |

**Table A2.** Symbol (column 1), description (column 2), value (column 3), and units (column 4) of parameters concerning the dimensions of the boxes used in our model.

| Symbol | Description | Value | Units |
|--------|-------------|-------|-------|
| $V_1$ | Volume of Box 1 | $2.71425 \times 10^{16}$ | $m^3$ |
| $V_2$ | Volume of Box 2 | $9.0475 \times 10^{15}$ | $m^3$ |
| $V_3$ | Volume of Box 3 | $2.442825 \times 10^{17}$ | $m^3$ |
| $V_4$ | Volume of Box 4 | $5.699925 \times 10^{17}$ | $m^3$ |
| $V_5$ | Volume of Box 5 | $4.523750 \times 10^{16}$ | $m^3$ |
| $V_6$ | Volume of Box 6 | $5.4285 \times 10^{17}$ | $m^3$ |
| $V_7$ | Volume of Box 7 | $9.0475 \times 10^{15}$ | $m^3$ |
| $A_1$ | Surface area Box 1 | $2.71425 \times 10^{14}$ | $m^2$ |
| $A_2$ | Surface area Box 2 | $3.619 \times 10^{13}$ | $m^2$ |
| $A_3$ | Surface area Box 3 | $2.71425 \times 10^{14}$ | $m^2$ |
| $A_4$ | Surface area Box 4 | $3.43805 \times 10^{14}$ | $m^2$ |
| $A_5$ | Surface area Box 5 | $1.8095 \times 10^{13}$ | $m^2$ |
| $A_6$ | Surface area Box 6 | $3.619 \times 10^{14}$ | $m^2$ |
| $A_7$ | Surface area Box 7 | $3.619 \times 10^{13}$ | $m^2$ |
| $d_{f1}$ | Floor depth Box 1 | 100 | m |
| $d_{f2}$ | Floor depth Box 2 | 250 | m |
| $d_{f3}$ | Floor depth Box 3 | 1000 | m |
| $d_{f4}$ | Floor depth Box 4 | 2500 | m |
| $d_{f5}$ | Floor depth Box 5 | 2500 | m |
| $d_{f6}$ | Floor depth Box 6 | 4000 | m |
| $d_{f7}$ | Floor depth Box 7 | 250 | m |
| $d_{c3}$ | Ceiling depth Box 3 | 100 | m |
| $d_{c4,1}$ | Ceiling depth Box 4 (below Boxes 2 and 7) | 250 | m |
| $d_{c4,2}$ | Ceiling depth Box 4 (below Box 3) | 1000 | m |
| $d_{c6}$ | Ceiling depth Box 6 | 2500 | m |

**Table A3.** Symbol (column 1), description (column 2), value (column 3), and units (column 4) of the other parameters used in our model.

| Symbol | Description | Value | Units |
|---|---|---|---|
| $Z_{1,base}$ | Base biological production Box 1 | 1.1 | mol C m$^{-2}$ yr$^{-1}$ |
| $Z_{2,base}$ | Base biological production Box 2 | 4.5 | mol C m$^{-2}$ yr$^{-1}$ |
| $Z_{5,base}$ | Base biological production Box 5 | 1.75 | mol C m$^{-2}$ yr$^{-1}$ |
| $Z_{7,base}$ | Base biological production Box 7 | 5.325 | mol C m$^{-2}$ yr$^{-1}$ |
| $\epsilon_{1,base}$ | Base biological efficiency Box 1 | 0.9 | - |
| $\epsilon_{2,base}$ | Base biological efficiency Box 2 | 1.25 | - |
| $\epsilon_{5,base}$ | Base biological efficiency Box 5 | 0.35 | - |
| $\epsilon_{7,base}$ | Base biological efficiency Box 7 | 0.62 | - |
| $T_{1,base}$ | Base temperature Box 1 | 23.34 | °C |
| $T_{2,base}$ | Base temperature Box 2 | 9.1 | °C |
| $T_3$ | Temperature Box 3 | 11.28 | °C |
| $T_4$ | Temperature Box 4 | 3.24 | °C |
| $T_{5,base}$ | Base temperature Box 5 | 0.93 | °C |
| $T_6$ | Temperature Box 6 | 1.8 | °C |
| $T_{7,base}$ | Base temperature Box 7 | 5.83 | °C |
| $S_1$ | Salinity Box 1 | 35.25 | g kg$^{-1}$ |
| $S_2$ | Salinity Box 2 | 34.27 | g kg$^{-1}$ |
| $S_3$ | Salinity Box 3 | 34.91 | g kg$^{-1}$ |
| $S_4$ | Salinity Box 4 | 34.76 | g kg$^{-1}$ |
| $S_5$ | Salinity Box 5 | 34.43 | g kg$^{-1}$ |
| $S_6$ | Salinity Box 6 | 34.77 | g kg$^{-1}$ |
| $S_7$ | Salinity Box 7 | 34.17 | g kg$^{-1}$ |
| $[Ca]_1$ | Calcium concentration Box 1 | 10.96 | mol m$^{-3}$ |
| $[Ca]_2$ | Calcium concentration Box 2 | 10.66 | mol m$^{-3}$ |
| $[Ca]_3$ | Calcium concentration Box 3 | 10.55 | mol m$^{-3}$ |
| $[Ca]_4$ | Calcium concentration Box 4 | 10.51 | mol m$^{-3}$ |
| $[Ca]_5$ | Calcium concentration Box 5 | 10.71 | mol m$^{-3}$ |
| $[Ca]_6$ | Calcium concentration Box 6 | 10.51 | mol m$^{-3}$ |
| $[Ca]_7$ | Calcium concentration Box 7 | 10.63 | mol m$^{-3}$ |

**Table A4.** The symbols and description of the equilibrium constants are presented in the first two columns. The third column presents the source of the used expression.

| Symbol | Description | Expression |
|---|---|---|
| $K_0$ | Solubility constant | Weiss (1974) |
| $K_1$ | First dissociation constant of carbonic acid | Lueker et al. (2000) |
| $K_2$ | Second dissociation constant of carbonic acid | Lueker et al. (2000) |
| $K_{sp,base}$ | Equilibrium constant for $CaCO_3$ dissolution | Mucci (1983) |
| $K_{sp,press}$ | Pressure correction for $K_{sp,base}$ | Millero (1983) |

*Author contributions.* DB and HD designed the study and constructed the AUTO version of the SPC-M. DB obtained and analysed all the results. DB and HD wrote the first version of the paper; all authors contributed to the writing of the final paper.

*Competing interests.* The authors declare that they have no conflict of interest.

*Acknowledgements.*

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
