# Peer review of "Effect of the Atlantic Meridional Overturning Circulation on Atmospheric pCO2 Variations"

_Earth System Dynamics, 2021_

## Author Comment (AC1)

**MS-No.:** ESD-2021-42

**Title:** Effect of the Atlantic Meridional Overturning Circulation on Atmospheric pCO$_2$ Variations

**Author(s):** Daan Boot, Anna S von der Heydt and Henk A. Dijkstra

**Point-by-point reply to reviewer #1**

**October 27, 2021**

We thank the reviewer for his/her careful reading and for the useful comments on the manuscript.

1. *The major issue relates to the parametrization added that gives rise to the internal oscillation. A part of this oscillation involves changes in the riverine flux of alkalinity as a function of pCO2 and the other is linked to an increase in temperature due to an increase in ocean alkalinity within 1000 years. What are the reasons behind these parametrizations? I understand that weathering is modulated by pCO2. However, I thought that this was a slow process, and I don't think that a change in atm. CO2 should directly lead to a proportional change in alkalinity river influx (within 1000 years). Maybe the oscillations you highight are relevant for longer timescales, i.e. glacial/interglacial changes in pCO2. I suggest to carefully read the litterature on changes in weathering during G-IG cycles. I can't find a reason for an increase in ocean alkalinity leading to an increase in temperature though (green box at t=0 to blue box at t=T/4 in fig. 6).*

   **Author's reply:**
   There are two different processes here: (1) the coupling between alkalinity and temperature, and (2) the riverine influx of alkalinity.

   (1) There is no direct coupling between alkalinity and atmospheric temperature. However, alkalinity indirectly influences temperature. It does this via its influence on the pH. The pH of the surface ocean determines the oceanic pCO$_2$. The gas exchange is proportional to the pCO$_2$ difference between the ocean and atmosphere. From this we see that the gas

exchange is influenced by the pH, and thus alkalinity. Via the gas exchange, atmospheric $pCO_2$ changes, and therefore also the atmospheric temperature.

Concerning figure 6: we understand the confusion here. In this figure, blocks of the same color have the same 'first event'. The first event is recognizable by the thick outlining. For the blue blocks this is 'Atmospheric pCO2 starts to increase'. This process continues for half a period, and is thus still present as alkalinity starts to increase.

(2) The parameterization used in this study is the same as used in the original SCP-M and is based on work by Toggweiler (2008).

It is true that riverine fluxes generally work on longer time scales (order 10 kyr). However, in the oscillation, our system does not reach equilibrium. The riverine influx is determined by atmospheric $pCO_2$ which again is influenced by processes on shorter timescales than the river fluxes. We would also like to point out that the amplitude of the river flux is small compared to that of the sinks of alkalinity in the oscillation (fig. 7b).

**Changes in manuscript:**
We will make Figure 6 clearer. Furthermore, we will clarify the role of the river flux in the oscillation mechanism.

2. *The paper is hard to follow. A combination of 13*7 experiments are performed. They are labelled with 1 or 2 letters per feedback and numbers for experiments, making it difficult to recall what we are looking at. If more explicit labels were used in Figures 3 and 4, it would help. In addition, there is very little justification/discussion of the different experiments, leading to confusion. The parametrization of the rain ratio feedback is not common. I thought that the largest impact on rain ratio would come from changes in silicifiers, and thus silicate and/or iron concentration in the ocean. L. 278, the authors state that "for low rain ratios, we only have a constant dissolution", which confuses me, as I don't see a link between dissolution and rain ratio in the methods.*

**Author's reply:**
We understand that the labelling of the experiments may be confusing. We will choose clearer, more explicit labels in the revision.

About the justification of the experiments: we will make it clearer in the

text. We generally choose experiments to test the effect of a feedback that is used in more complicated models. Feedbacks that were more certain (such as the temperature feedback ($\lambda_T > 0$)) or had a large effect on the solution (such as the efficiency feedback ($\lambda_\epsilon > 0$)) were used in experiments with more than one feedback.

The parameterization of the rain ratio feedback used in this study is taken from Ridgwell et al. (2007). This parameterization is also optional in the EMIC CSIRO Mk3L-COAL model (Buchanan et al., 2019). Our model does not include silicifiers and/or iron. Therefore, we do take these effects into account.

In our model, dissolution of $CaCO_3$ is dependent on two components: (1) a component proportional to the rain ratio and related to the saturation state; and (2) a constant component. When the saturation state is larger than 1, the first component is equal to 0. For this specific experiment the saturation state is always above 1 when the rain ratio is low. So what is meant with L. 278 is that when the rain ratio is low, the saturation state is always larger than 1, thus we have no saturation driven dissolution, but only a constant dissolution (the second component).

**Changes in manuscript:**
We will use more explicit labels for the experiments. Furthermore, we will include a better justification for the performed experiments and we will make the text around L.278 clearer.

3. *Discussion and implication of the results:*
   *The study scans a large range of parameters yielding pCO2 values of 70-300 ppm, but without really trying to assess physical plausability. For example, in Figure 4, multipliers 0.1-10 are included in the parametrizations, but without much justification. What can the authors deduce from their results? What are the probable ranges?*

   *The discussion needs to put the results back in context and discuss them in light of previous experiments. In the Introduction, the authors cite previous studies that simulated the impact of AMOC changes on the carbon cycle with Earth system models (in which most of the feedbacks explored were included). Can your results help understand better these previous simulations?*

**Author's reply:**
We agree. Reviewer 2 also commented on the justification of these experiments.

**Changes in manuscript:**
We will include more justification of these experiments and discussion of the results.

Minor points:

1. *L. 41: I am not sure that "not well understood" is appropriate, since a lot of studies have highlighted the impact of AMOC on pCO2 and the reverse as highlighted in the 2 following paragraphs. It is however a complex interaction.*

   **Author's reply:**
   We agree that it is not the most appropriate wording.

   **Changes in manuscript:**
   We will change the text to reflect the complex interactions.

2. *L 272: Please amend: "Fig. 4a, b is yellow.."*

   **Author's reply:**
   Suggestion followed

   **Changes in manuscript:**
   The text will be changed accordingly.

3. *L. 295: What is the meaning of "we continue in the piston velocity"?*

   **Author's reply:**
   This means that we use the piston velocity parameter as our continuation parameter. **Changes in manuscript:**
   We will clarify this in the revised text.

---

## Author Comment (AC2)

**MS-No.:** ESD-2021-42

**Title:** Effect of the Atlantic Meridional Overturning Circulation on Atmospheric pCO$_2$ Variations

**Author(s):** Daan Boot, Anna S von der Heydt and Henk A. Dijkstra

**Point-by-point reply to reviewer #2**

**October 27, 2021**

We thank the reviewer for his/her careful reading and for the useful comments on the manuscript.

1. *In the Inroduction section, are there any papers using 3D OGCM to simulate the atmospheric pCO2-AMOC strength relationship under PI and LGM? If so, these papers need to be properly cited.*

   **Author's reply:**
   In the introduction we already cite multiple papers that simulate the atmospheric pCO$_2$-AMOC relationship using EMICs, ESMs and (A)OGCMs.

   Examples of the cited papers are:
   Menviel et al. (2014) (EMIC in the LGM); Menviel et al. (2008) (EMIC in the LGM and PI); Mariotti et al. (2012) (AOGCM in the LGM); Nielsen et al. (2019) (ESM with a PI control simulation); Huiskamp and Meissner (2012) (ESM in the LGM); Gregory et al. (2005) (AOGCMs and EMICs with a PI control simulation); and Gottschalk et al. (2019) (ESMs and EMICS in the LGM).

   However, we may have missed some interesting papers.

   **Changes in manuscript:**
   We will look into the literature again, and add new citations to the introduction.

2. *I didn't see any experiments to test the plausibility of the box model to address the AMOC-pCO2 relationship problem. I would suggest that you set up two more experiments fully including all the feedbacks you mentioned in Table 2 and check if the atmospheric pCO2 is reasonable under two scenarios.*

**Author's reply:**
That is a good suggestion.

**Changes in manuscript:**
Suggestion will be followed. We will include results of these two extra experiments.

3. *In general, I think all the experiments should be set up with other feedbacks properly included to make the case more realistic. For example, when studying the role of biological feedback (x-0 and x-1 in Table 2), the x-0 could be set up with all $\lambda = 1$, x-1 then should be only with $\lambda_{BI}=0$, etc.*

   **Author's reply:**
   We chose to set up the experiments as in the original paper, since we base our model on the SCP-M and this model contains no feedbacks. The SCP-M is tuned to accurately represent both the PI and LGM conditions. We therefore consider that we start with a "realistic model" if all feedbacks are switched off (i.e. experiment x-0). Switching on all the feedbacks would not necessarily lead to a more realistic case, since the SCP-M is not tuned to include these parameters.

   **Changes in manuscript:**
   We will better justify our approach.

4. *In lines 266-270, the three parameters are selected as control parameters: the rain ratio, the biological production and the piston velocity. Please explain the reasons for picking these parameters. Also, the multiplier changes from 0.1 to 10 without reasonable explanations. I would suggest using more realistic ranges.*

   **Author's reply:**
   We use these three parameters since they more or less represent the three carbon pumps often used in the traditional view of the oceanic carbon cycle. The rain ratio affects the strength of the carbonate pump, the biological production the soft tissue pump and the piston velocity the solubility pump. We chose these three parameters to see whether a (large) change in one of the traditional pumps can invoke large non-linear changes or bifurcations in this model.

We agree that the multiplier range does not necessarily reflect realistic values. One of the goals of this study was to get a better understanding of the sensitivity of the carbon cycle to these parameter changes and whether bifurcations can arise.

**Changes in manuscript:**
We will better motivate the reasons for varying these three parameters, and why we choose a large range in parameter values.

Comments/concerns about specific feedbacks/parameters are below.

1. *In equation (2), the authors chose 0.54 °C/(Wm-2) to compute the temperature change. As this parameter is important in equation (12) to control the AMOC strength, what is the sensitivity of this parameter to coupling AMOC-carbon cycle?*

   **Author's reply:**
   The precise value of this parameter (0.54) is not very important in this study as the sensitivity to this parameter is generally low. This can also be seen in section 3.2., where we check the sensitivity of atmospheric $pCO_2$ to the value of $\lambda_A$. This can also be interpreted as the sensitivity of the relationship to this 0.54 (since 0.54 is multiplied with $\lambda_A$), which is low.

   We do see that the system is prone to show Hopf bifurcations when $\lambda_T$ is increased. However, this is when we increase $\lambda_T$ to relatively large values (order 20).

   **Changes in manuscript:**
   No changes necessary.

---

## Editor Decision (ED1)

The authors have adequately addressed most of the reviewers' comments.

There is one comment that remains to be addressed, which relates to the timing of the riverine alkalinity flux. As reviewer #1 notes in their first-round review comments, weathering is a slow process, and they question that changes in atmospheric CO2 affect the riverine flux on a millennial timescale. In their response 2(b) to reviewer #2 comments, the author provide considerations about timing, but these are unclear and were not reflected in the manuscript. Please clarify your response on the timescale of the riverine flux, and revise the manuscript accordingly.

On the revised manuscript, parameterizations for the riverine flux and CaCO3 dissolution are provided in the results, rather than in the model description section of paper. Please consider if they would be better placed in the model description section. E.g. the riverine flux parameterization could be described in the first paragraph on p. 5, where the flux is first introduced.

Please read the manuscript carefully before submission as a few typos were introduced in the revision.

---

## Author Response (AR2)

Dear Kirsten Zickfeld,

Thank you for the effort you have put in our manuscript so far, for which we have uploaded a revised version.

Our apologies for not sufficiently answering the points raised by the second reviewer in the first revision. We have addressed the points raised by the second reviewer in much detail for this revision. We kindly ask you if you can approach the second reviewer again to ask whether the reviewer wants to have another look at our revised paper.

Thank you in advance.

Kind regards,
Daan Boot
On behalf of all authors.

**MS-No.:** ESD-2021-42

**Title:** Effect of the Atlantic Meridional Overturning Circulation on Atmospheric pCO$_2$ Variations

**Author(s):** Daan Boot, Anna S von der Heydt and Henk A. Dijkstra

**Point-by-point reply to reviewer #1**

**March 23, 2022**

We thank the reviewer for his/her careful reading and for the useful comments on the manuscript.

Minor revisions:

1. *In the introduction (L41-L52), why not specify the sign of atmospheric pCO$_2$ changes via each mechanism?*

   **Author's reply:**
   We have not specified the signs of the mechanisms because the signs of the mechanisms are not necessarily the same for each study. Some studies simulate pCO$_2$ decrease and some simulate pCO$_2$ increases after an AMOC weakening. This is addressed in L49-L52, i.e. the difference in model complexity, model time scales, and climatic boundary conditions are the reasons behind this.

   **Changes in manuscript:**
   No changes necessary.

2. *Around L300-314: As the biological productivity increases (Figure 4c and 4d: fixed rain ratio), both the CaCO$_3$ pump and soft tissue pump change. I didn't see the explanation of changing soft-tissue pump on atmospheric pCO$_2$. Does the contribution of CaCO$_3$ mechanism over dominates the changing soft-tissue pump? I would suggest to clarify it in the text.*

   **Author's reply:**
   Generally, one would expect that a weaker soft tissue pump (STP) results in lower carbon export to the deep ocean, and therefore higher concentrations in the surface ocean leading to higher pCO$_2$ values. This

is what occurs in the range $0.7 - 1.0$ of the biological productivity parameter and when this parameter is lower, $pCO_2$ is higher. As biological production becomes weaker, $CaCO_3$ production and burial also decrease. If burial decreases, the river influx has to decrease to conserve alkalinity and therefore atmospheric $pCO_2$ has to decrease. This is what happens when the biological productivity parameter is smaller than $0.7$.

**Changes in manuscript:**
A more extensive discussion of this process is included in the main text (paragraph around line 300).

3. *In Table 2, how do you decide the parameter in x-HB experiment?*

   **Author's reply:**
   The combination of SCP-M and AUTO makes exploring the parameter space very cheap. This allowed us to do an extensive scanning of the parameter space where we eventually discovered the Hopf Bifurcation (HB). We explored the existence of the HB in the parameter space and finally settled on the parameter values presented in the paper as we thought these parameter values are within a likely realistic range.

   **Changes in manuscript:**
   This is clarified in the main text (around L. 315).

4. *In Figure 2, in the LGM configuration, does the HB exist in the experiment of L-CTL and L-BIO? If so, does the atmospheric $pCO_2$ mean the unstable FP $pCO_2$?*

   **Author's reply:**
   No, the HB does not exist. Atmospheric $pCO_2$ in Figure 2 represents a stable fixed point.

   **Changes in manuscript:**
   This is clarified in the main text (around L. 240) and in the caption.

5. *Does HB exist or is it possible to find the HB using your model if the $pCO_2$-AMOC feedback is included?*

   **Author's reply:**
   We find the HB when we use the AMOC as control parameter. When we enable the $pCO_2$-AMOC feedback, we cannot use the AMOC strength as a control parameter anymore. Given that atmospheric $pCO_2$ is not

very sensitive to changes in the AMOC, we can expect the strength of this feedback to be relatively small as the results in section 3.3 also suggest. So, we cannot say it with certainty, but we do expect it is likely that the HB exists in the parameter space when the $pCO_2$-AMOC feedback is enabled

**Changes in manuscript:**
We address this in the revised discussion of the results (around L.420).

**MS-No.:** ESD-2021-42

**Title:** Effect of the Atlantic Meridional Overturning Circulation on Atmospheric pCO$_2$ Variations

**Author(s):** Daan Boot, Anna S von der Heydt and Henk A. Dijkstra

**Point-by-point reply to reviewer #2**

**March 23, 2022**

We thank the reviewer for his/her careful reading and for the useful comments on the manuscript.

*This is my second review of Boot et al. Unfortunately, I don't think my comments have been adequately addressed by the authors. I suggested Major revisions, but only a few minor changes were made to the manuscript.*

1. *The relationship between atmospheric $CO_2$ and riverine influx is still not clear (L. 359-364 does not explain that relationship). Line 334-335 it is stated that the river influx is directly proportional to pCO$_2$ and it starts to change at t=0 in Fig. 6. This link between $CO_2$ and river influx has to be explained and justified (both in terms of amplitude and timing) in a much better way in the manuscript. The assumptions taken (since it is a very much simplified framework) have to be clearly spelt out and discussed.*

   *Similarly, in a more complex system, $CaCO_3$ burial would not be a simple function of $CaCO_3$ production at the surface (L. 335-336). While Figure 6 focuses on the relationship between alkalinity and air-sea $CO_2$ exchange, changes in biological efficiency and $CaCO_3$ production will also impact DIC and thus air-sea gas exchange and SST will impact $CO_2$ solubility.*

   ***Author's reply:***
   *In the above paragraph multiple points are raised and therefore we have divided this reply in multiple sections: 1. River flux: (a) What is the relation between the river flux and atmospheric pCO$_2$? (b) Where does this relation come from? (c) What assumptions underlie this relation? (d) How does this choice influence our results? 2. $CaCO_3$ burial: (a)*

*What is the $CaCO_3$ burial relation? (b) Where does it come from? (c) Why is it only dependent on $CaCO_3$ production? 3. Other processes: What is the role of other processes such as the impact on DIC and solubility of $CO_2$?*

*1. (a)* **What is the relation between the river flux and atmospheric $pCO_2$?** *The relation for carbon influx is given by:*

$$C_{river} = W_{SC} + (W_{SV} + W_{CV}) \times pCO_2^{atm} \tag{1}$$

*Where $W_{SC}$ is a parameter reflecting constant silicate wheatering, $W_{SV}$ a parameter representing variable silicate weathering, and $W_{CV}$ a parameter representing variable carbonate weathering. $pCO_2^{atm}$ represents the partial pressure of $CO_2$ in the atmosphere. This relation comes from O'Neill et al. (2019) which is directly taken from Toggweiler (2008). No adaptations have been made to the relation, nor the parameter values.*

*The parameter values are: $W_{SC} = 0.75 \times 10^{-4}$ mol $m^{-3}$ $yr^{-1}$; $W_{SV} = 0.50 \times 10^{-4}$ mol $m^{-3}$ $atm^{-1}$ $CO_2$ $yr^{-1}$; and $W_{CV} = 2.00 \times 10^{-4}$ mol $m^{-3}$ $atm^{-1}$ $CO_2$ $yr^{-1}$*

*1. (b)* **Where does this relation come from?** *The relation used in Toggweiler (2008) is based on the relation used in Walker and Kasting (1992), i.e.,*

$$C_{river} = W_{SV} \times (pCO_2^{atm})^{0.3} + W_{CV} \times pCO_2^{atm} \tag{2}$$

*where $W_{CV}$ is $0.00015 \times 10^{17}$ mol $yr^{-1}$ and $W_{SV}$ is $0.00005 \times 10^{17}$ mol $yr^{-1}$, or when the volume of box 1 in the SCP-M is taken into account $5.7 \times 10^{-4}$ mol $m^{-3}$ $yr^{-1}$ and $1.9 \times 10^{-4}$ mol $m^{-3}$ $yr^{-1}$. The main addition in Toggweiler (2008) is the constant silicate weathering and the change from a power law to a linear relationship.*

*In Walker and Kasting (1992) they state that the key element in the representation in continental weathering rate is how the carbonate and silicate dissolution rates depend on $pCO_2$. In the long term (100,000s of years), this dependence determines steady state $pCO_2$. On shorter time scales (10s-100s of years), it is this dependence that determines how quickly the ocean-atmosphere system will recover from an impulsive*

*input of fossil fuel carbon dioxide. They mention that several formulations have been proposed, but that all of those rate laws are subject to large uncertainty. There are dependencies on the hydrological cycle, on global average temperature, but for simplicity they express the weathering rate as a function $pCO_2$. Since the weathering rate law controls $CO_2$ response on a time scale of 100,000s to 1,000,000s of years, they state it is not that important which formulation is used as long as processes on time scales shorter than that are studied.*

*The parameter values are chosen such that the influx via the rivers is approximately equal to estimates of carbonate influx via rivers (e.g. in Milliman and Droxler, 1993).*

*1. (c) **What assumptions underlie this relation?** The main assumptions are:*

- *For the model complexity and time scales addressed,the relation to atmospheric $pCO_2$ is sufficiently realistic.*

- *River influx occurs only in the low latitude ocean ($30^\circ S - 30^\circ N$).*

- *River influx is due to carbonate and silicate continental weathering.*

- *Silicate weathering consists of a constant and variable part.*

- *Carbonate weathering consists of only a variable part.*

- *There is no delay between continental weathering and river influx.*

- *Parameter values are insensitive to changes between LGM, Holocene and Anthropocene.*

*1. (d) **How does this choice influence our results?** The parameter values, and thus the amplitude of the river flux are important for the amplitude of the change in total alkalinity in the system, and the alkalinity concentration in the surface ocean. Decreasing the amplitude of the river flux, would decrease the amplitude of total alkalinity in the ocean and thus the amplitude of the $CO_2$ oscillation in the atmosphere. Decreasing the amplitude enough might make the oscillation disappear. Increasing the amplitude would result in unphysical results, since atmospheric $pCO_2$ values would reach too low values, decreasing ocean temperatures below freezing point. Therefore, the currently used*

*parameter values, based on estimated carbonate input via rivers, are in a range such that the Hopf bifurcation (HB) exists.*

*The fact that there is no time delay between atmospheric $pCO_2$ and the river influx is important. The approximate quarter period delay between atmospheric $pCO_2$ and total alkalinity (and alkalinity in box 1) is important for driving the oscillation. The river influx plays a role in this by changing the alkalinity in box 1. However, only if the delay is multiple centuries, we expect very different results than the ones presented here.*

*To conclude: (1) The assumption of the coupling of atmospheric $pCO_2$ to the river influx is expected not to be crucial. (2) The magnitude of the parameter values is important, but the actual magnitude of the parameter values are reasonable. (3) The fact that there is no delay between $pCO_2$ and the river influx is important, but the current assumption is reasonable for the model complexity and time scales involved.*

*Other assumptions, such as the division of the river influx in carbonate and silicate weathering are common in these types of models and are not expected to influence the results significantly. The fact that we do not couple the river influx to different processes such as the hydrological cycle, is related to the arguments of the model complexity and time scales assessed.*

*2. (a) **What is the $CaCO_3$ burial relation?** The change in DIC due to $CaCO_3$ related processes is dependent on three different processes: (1) Formation of $CaCO_3$ (sink of DIC in surface boxes); (2) Dissolution of $CaCO_3$ in the water column (source of DIC in all boxes); (3) Dissolution of $CaCO_3$ in the sediments (source of DIC in abyssal box). Burial of $CaCO_3$ is defined as the difference between the formation (process 1) and dissolution of $CaCO_3$ (processes 2 and 3).*

*Process 1 varies because the formation of $CaCO_3$ is dependent on biological production which is variable due to changes in phosphate upwelling and biological efficiency. Processes 2 and 3 are based on the same general relation:*

$$C_{Diss} = ([CO_3^{2-}][Ca^{2+}]) \times k_{Ca} \times (1 - (min(([\frac{[CO_3^{2-}][Ca^{2+}]}{K_{sp}}), 1)))) + D_C$$
$$(3)$$

*The first part is related to the saturation state:* $\frac{[CO_3^{2-}][Ca^{2+}]}{K_{sp}}$. *If the saturation state is larger than 1, the saturation dependent dissolution is 0, and only the constant term remains ($D_C$).*

*In the oscillation, the saturation state of $CaCO_3$ in the ocean is everywhere larger than 1. Therefore, total dissolution in the ocean is constant and does not vary. This results in that the $CaCO_3$ burial becomes a function of $CaCO_3$ formation and thus biological production.*

*2. (b) **Where does it come from?** The saturation driven part is a very general formulation used to describe $CaCO_3$ dissolution (e.g. Sarmiento and Gruber, 2006). Observations suggest that even in supersaturated cases some form of dissolution occurs (Harrison et al., 1993; Milliman et al., 1999) which is the reason for including a constant dissolution term.*

*2. (c) **Why is it only dependent on $CaCO_3$ production?** Using a simple model for $CaCO_3$ dissolution, Zeebe and Westbroek (2003) show that the entire ocean can be supersaturated in $CaCO_3$ when the riverine influx is larger than the biogenic flux in the surface ocean. In the SCP-M, the river influx is indeed larger than biological production, representing the situation sketched in Zeebe and Westbroek (2003). In the current ocean and the LGM ocean, this situation is not likely, but this does not mean it has not happened in the past.*

*3. **What is the role of other processes such as the impact on DIC and solubility of $CO_2$?** The processes described in the main text are the driving forces behind the oscillation. Other processes are also important in shaping the period and amplitude of the oscillation, since the processes are all dependent on each other. Generally the change in DIC, Alk and $PO_4^{3-}$ concentrations is due to a balance of several large fluxes that are sometimes more than 100 times larger than the sum of all fluxes. It is therefore difficult to describe the effect of each process individually. However, the coupling between atmospheric $pCO_2$ and the alkalinity cycle appears to be the driving mechanism of this oscillation.*

***Changes in manuscript:***
*1. (a) This relation and the parameter values are included in the main text (Eq. 13 and surrounding text). 1. (b) This is discussed in the main text (Eq. 13 and surrounding text). 1. (c) This is addressed*

*in the discussion in combination with 1(d) (paragraphs around L. 440-465). 1. (d) This is addressed in the discussion (paragraphs around L. 440-465). 2. (a) The relation is clarified in the text (Eq. 14 and surrounding text). 2. (b) No changes necessary. 2. (c) We mentioned this in the main text (paragraph around L. 350). 3. We included a more thorough discussion in the main text (paragraph around L. 367-376).*

2. *A wide range of parameter space and their impact on $pCO_2$ is explored, however there is little explanation as to why this would occur, if this is plausible in a more complex system and the implications of the results are not discussed. The authors need to discuss how their results can help in better understanding climate-carbon cycle interactions. They also need to discuss the assumptions taken, their limits and how this might impact the results.*

   *Another aspect to discuss is the fact that the current study focuses on box 1 (low latitude surface box), whereas most studies suggest high latitude processes dominate the carbon cycle.*

   **Author's reply:**
   *This reply is also divided based on the multiple subcomments: (a) Why is such a wide range of parameter space explored? (b) Is this possible in more complex systems? (c) How do the results help in better understanding climate-carbon cycle interactions? (d) What assumptions did we make, what are the limitations of the model, and how does this impact our results? (e) Why is there a contrast between our study where low latitude processes seem to be important, while other studies suggest high latitude processes to dominate the carbon cycle?*

   *(a) **Why is such a wide range of parameter space explored?** In this study there are a few parameters we vary: AMOC strength, the rain ratio, biological production, the piston velocity and climate sensitivity.*

   *First of all, we have varied these parameters over a wide range to look for multiple (stable) equilibria. By following branches of steady state solutions, we were curious whether we would find saddle-node bifurcations. Saddle node bifurcations might exist for unlikely parameter values, but this would show that there are multiple stable equilibria, possibly in the more realistic parameter value ranges. We did not find any saddle node bifurcations in this model, suggesting that in this simple carbon cycle box model, no multiple stable equilibria exist. This is*

*valuable information from a dynamical systems point of view.*

*Secondly, a large part of our research focuses on changes in the AMOC strength. Multiple studies have shown that the AMOC can have multiple stable equilibria. Disruption of the AMOC is often thought to be a cause for major climate shifts such as the Dansgaard-Oeschger events. Future projections show that the AMOC strength is decreasing significantly (e.g. 48% in SSP5-8.5 using the CESM2 model in a concentration driven simulation). By varying the AMOC strength, we see how the carbon cycle responds to changes in this AMOC, and whether a possibly weaker AMOC in the future might impact atmospheric $pCO_2$.*

*We varied the rain ratio, biological production and piston velocity to assess the sensitivity of atmospheric $pCO_2$ to changes in one of the three traditional carbon pumps (carbonate, soft tissue, solubility, respectively). The ranges for these parameters are larger than realistic, mostly because we were also testing to see whether there are bifurcations in the model. The results show the sensitivities in this model to changes in these model parameters and in the more realistic parameter ranges, interesting results are found.*

*We also varied climate sensitivity. Climate sensitivity of the Earth is still uncertain and Earth System Models give a wide range (1.8-5.6 K in CMIP6; Zelinka et al., 2020). By using multiple values of this climate sensitivity, we take this uncertainty into account.*

*Lastly, by using these wide ranges, we have covered most of the realistic response in this model. We also covered some unrealistic ranges, but the main take home message is: the marine carbon cycle (in this model) seems to be a stable system where no large changes occur when parameters are varied.*

*(b)* **Is this possible in more complex systems?** *Not all combinations of values are possible in more complex systems. A realistic range for the AMOC strength is a few Sverdrups to approximately 25 Sv, which mean most of our results are in a realistic AMOC range. Climate sensitivity is also within realistic ranges in our study. The three parameters representing the different pumps are mostly outside realistic ranges, but cover the realistic ranges.*

*(c)* **How do the results help in better understanding climate-carbon cycle interactions?** *We have shown that the carbon cycle*

*is quite a stable system with no saddle node bifurcations in this simple model. This suggests that as the climate changes, we do not expect to find so-called tipping points in the carbon cycle as a response to climate change. Furthermore, we show that changes in AMOC strength, which are very likely under changing climate and have happened in the past, do not result in large changes in atmospheric $pCO_2$. Lastly, if we want to explain variability in the past climate linked to the carbon cycle, oscillations due to Hopf bifurcations should also be taken into account.*

**(d) **What assumptions did we make, what are the limitations of the model, and how does this impact our results?**

*One of the assumptions we made is that the original SCP-M performs well for Last Glacial Maximum and Pre-Industrial conditions. In O'Neill et al. (2019) they discuss this, and the model gives reliable results for these two time periods. We have added several features to the model and one of our assumptions is that the parameter values do not need to be retuned after the addition of these new features. We also assume that the simplification of the carbonate chemistry does not change the outcome significantly. Other minor assumptions we made are that the steady state $AMOC\text{-}pCO_2$ relation can be captured by a logarithmic function, and that the temperature effect on biological efficiency is a linear function.*

*Limitations of the original SCP-M include: there is no distinction between ocean basins, which means the framework may not be useful for testing localized or detailed problems; and there is a rigid and somewhat arbitrary treatment of box boundaries. In our application, we also consider the simple carbonate chemistry as a limitation.*

*The impact on the results differ per assumption and limitation. Since the SCP-M performs well in the original paper for two different cases (O'Neill et al., 2019). By adding extra features to the model, some refitting of parameters was necessary, depending on the specific (newly added) processes: (i) The efficiency parameters in the biological feedback have been fitted to give similar results as the original model. (ii) The feedbacks related to the temperature, biological efficiency, piston velocity and rain ratio are not expected to result in necessary refitting of the parameter values. This is because the temperature feedback does not directly affect any parameters in the elemental cycles, and the other*

*feedbacks do not add new processes to the model. Parameters are just replaced by more complex, variable functions, but values remain close to the original values. (iii) The addition of the biological alkalinity flux might make refitting of parameters necessary since a complete new process is added to an elemental cycle. This would be a large exercise and would also make comparison between the different cases difficult. This means that cases where this feedback is used should be approached more carefully. However, in our results without refitting, cases with this feedback do not show divergent results compared to other cases.*

*An important change we made is the simplification of the carbonate chemistry. This change typically reduces pH by 0.15-0.2 (Munhoven, 2013). Another effect of this simpler chemistry is that atmospheric $pCO_2$ can have changes of the order of 20% (Munhoven, 1997; Munhoven, 2013), which explains the approximate 60 ppm lower atmospheric $pCO_2$ in our model compared to the original version. Obviously this assumption impacts our results, since atmospheric $pCO_2$ values are clearly affected. However, general sensitivities as discussed in our results are not expected to change a lot.*

*Our results show that atmospheric $pCO_2$ is not very sensitive to changes in the AMOC, or to the strength of the AMOC-$pCO_2$ feedback we have used. Therefore, we expect that different formulations of this feedback do not change the results significantly. The assumption that biological efficiency is linearly related to change in temperature is very uncertain, while the biological efficiency feedback is important for the oscillation we found. What seems to be important for the oscillation is that the coupling between atmospheric $pCO_2$ and the biological efficiency is strong enough. Whether the strong relation chosen now is realistic is uncertain.*

*One of the limitations of the original SCP-M is that there is no distinction between ocean basins. This might have an impact via the northern high latitude box (box 2), since the circulation in this box represents the AMOC, which is of course not present in the Pacific. In the original model, global values were still representative, showing that it might not have a large impact on the results. The impact of the other limitations on the results are hard to determine (box boundaries) or already discussed (carbonate chemistry).*

*(e)* **Why is there a contrast between our study where low latitude processes seem to be important, while other studies suggest high latitude processes dominate the carbon cycle?**

*It seems that the existence of box 1 is important in the oscillation which suggests that low latitude processes are important. However, the oscillation is not necessarily driven by low latitude processes. It is a global process, i.e. $CaCO_3$ production over the entire ocean decreases, and not just in the low latitude surface box (box 1). The role of box 1 is important since the air-sea gas exchange in box 1 leads atmospheric $pCO_2$.*

**Changes in manuscript:**
*(a) This is addressed in the discussion (paragraph around L.465-475).*
*(b) This is addressed in the discussion (paragraph around L.465-475).*
*(c) This is addressed in the discussion (last paragraph section 4; around L475-482). (d) This is addressed in the discussion (paragraphs around L. 440-465). (e) This is addressed in the result section (end of section 3.3; around L. 390).*

*Minor points:*

1. *L. 262-263: Why would deep ocean ventilation be more sensitive to AMOC changes when the AMOC is weak?*

   **Author's reply:**
   *If we see deep ocean ventilation as the sum of the General Overturning Circulation (GOC; $\psi_1$) and the AMOC ($\psi_2$) then this sum is relatively more sensitive to changes in the AMOC as the GOC is lower (in the LGM case).*

   **Changes in manuscript:**
   *This is clarified in the text (around L. 260).*

**MS-No.:** ESD-2021-42

**Title:** Effect of the Atlantic Meridional Overturning Circulation on Atmospheric pCO$_2$ Variations

**Author(s):** Daan Boot, Anna S von der Heydt and Henk A. Dijkstra

**List of changes second revision**

**March 23, 2022**

Following is a list of notable changes made in this second revision as a response to the comments of the reviewers. There have also been some small textual changes (spelling, rephrasing, etc.). These small changes are not included in this list.

1. We have extended the explanation of the two regimes in Section 3.2 where we use biological productivity as control parameter (paragraphs around L. 300).

2. We have included a more thorough discussion on the role of the river influx and CaCO$_3$ burial in Section 3.3 (text around Eq. 13 and Eq. 14).

3. We have included a paragraph discussing how other processes influence the oscillation (paragraph around L. 370).

4. We have extended Section 4 with a discussion on the assumptions, limitations and impact of our study (from L. 440 to the end).

---

## Author Response (AR3)

**Changes made to the manuscript:**
- We have rewritten and clarified the part of the discussion regarding the timing of the river flux and the fact that the flux responds instantaneously to atmospheric $pCO_2$ variations.
- As suggested by the editor we moved the equations of the river flux and dissolution to the model description section.
- We have corrected some typos.

---

## Author Response (AR4)

**Author response**

*'The reviewer's comment about the timing of the riverine alkalinity flux has still not been adequately addressed. As the authors note, the assumption that the flux responds immediately to changes in atmospheric CO2 is central to the mechanism generating the oscillation, yet a justification for this assumption is not given. Weathering is a slow process (100,000 yr timescale) and an explanation needs to be provided as to how it can affect the riverine flux on a millennial timescale.'*

**Author reply**

To completely respond to these comments, we will clarify what we have written in the main text and how the weathering is parameterized in the model.

First of all, the size of the river influx responds to changes in pCO2 immediately, but this does not mean it has a significant impact on ocean carbon and alkalinity. The timescale the river influx becomes important in this model is the timescale of the oscillation.

The river flux parameterization consists of two different processes: carbonate weathering and silicate weathering, where silicate weathering consists of a variable and constant part and carbonate weathering only of a variable part. Note that only the variable weathering components are important for the oscillation. Looking at the parameter values, carbonate weathering is responsible for 80% of this variability and silicate weathering only 20%.

These two different processes act on different timescales. The reviewer refers to weathering on $10^5$ years timescales. This refers to silicate weathering. Silicate weathering balances the volcanic input of carbon on the $10^5$ to $10^6$ years timescales, timescales order larger than the timescale of the oscillation. However, terrestrial carbonate weathering is important on shorter timescales: $10^3$ to $10^4$ years, the timescale of the oscillation found in this study. These timescales have been found in multiple studies (e.g. Archer et al., 1997; Lenton and Britton, 2006; Sarmiento and Gruber, 2006; Brault et al., 2017). We therefore want to stress that the weathering important for the oscillation is carbonate weathering and not silicate weathering.

Lastly, we want to repeat that our system does not reach a steady state and that the amplitude of the river influx is more than two times smaller than the burial of $CaCO_3$ in the ocean and therefore less important.

**References:**

- Archer, D. [E.], H. Kheshgi, and E. Maier-Reimer (1997), Multiple timescales for the neutralization of fossil fuel CO2, Geophys. Res. Lett., 24, 405–408.
- Lenton, T. M. and Britton, C.: Enhanced carbonate and silicate weathering accelerates recovery from fossil fuel CO2 perturbations, Global Biogeochem. Cy., 20, GB3009, https://doi.org/10.1029/2005GB002678, 2006.
- Sarmiento, J. and Gruber, N.: Ocean Biogeochemical Dynamics, Princeton University Press, Princeton, 2006.
- Brault, M.-O., Matthews, H. D., and Mysak, L. A.: The importance of terrestrial weathering changes in multimillennial recovery of the global carbon cycle: a twodimensional     perspective,     Earth     Syst.     Dynam.,     8,     455–475, https://doi.org/10.5194/esd-8-455-2017, 2017.

**Changes in manuscript**

In the description of the river flux parameterization, the difference between silicate and carbonate weathering has been made more explicit and the paragraphs around line 390 and line 450 are clarified and extended to reflect the response above.